# A tissue boundary orchestrates the segregation of inner ear sensory organs

**Ziqi Chen†, Magdalena Żak*†, Shuting Xu, Javier de Andrés, Nicolas Daudet**

UCL Ear Institute, University College London, London, United Kingdom

## eLife Assessment

This is an **important** study describing the morphological changes during boundary formation between sensory and non-sensory tissues of the inner ear. The authors provided **solid** evidence that a transcription factor, Lmx1a and ROCK-dependent actinomyosin are key for border formation in the inner ear. However, future studies will be needed to investigate the direct relationships among boundary formation, Lmx1a and ROCK. This work will be of interest to developmental biologists interested in boundary formation.

**\*For correspondence:**
magdalena.zak1@gmail.com

†These authors contributed equally to this work

**Competing interest:** The authors declare that no competing interests exist.

## Abstract

The inner ear contains distinct sensory organs, produced sequentially by segregation from a large sensory-competent domain in the developing otic vesicle. To understand the mechanistic basis of this process, we investigated the changes in prosensory cell patterning, proliferation, and character during the segregation of some of the vestibular organs in the mouse and chicken otic vesicle. We discovered a specialised boundary domain, located at the interface of segregating organs. It is composed of prosensory cells that gradually enlarge, elongate, and are ultimately diverted from a prosensory fate. Strikingly, the boundary cells align their apical borders and constrict basally at the interface of cells expressing or not the Lmx1a transcription factor, an orthologue of *Drosophila* Apterous. The boundary domain is absent in *Lmx1a*-deficient mice, which exhibit defects in sensory organ segregation and is disrupted by the inhibition of ROCK-dependent actomyosin contractility. Altogether, our results suggest that actomyosin-dependent tissue boundaries ensure the proper separation of inner ear sensory organs and uncover striking homologies between this process and the compartmentalisation of the *Drosophila* wing disc by lineage-restricted boundaries.

## Introduction

The inner ear of vertebrates is composed of a series of liquid-filled chambers containing distinct sensory organs. Each organ contains a sensory epithelium populated with mechanosensory 'hair cells', separated from one another by supporting cells. In amniotes, the ventral part of the inner ear forms the cochlea, which hosts an auditory epithelium called the basilar papilla in birds and crocodiles, or the organ of Corti in mammals. The dorsal part forms the vestibular system, with three cristae (anterior, posterior, and lateral) and their associated semicircular canals responding to the angular rotations of the head, and the maculae of the utricle and saccule, sensitive to gravity and linear acceleration. These sensory organs are separated from one another by non-sensory territories with essential roles in inner ear function and homeostasis (*Ekdale, 2016*).

The development of the inner ear is a remarkable example of 3D tissue morphogenesis that starts with the formation of the otic placode, an epithelial thickening located on both sides of the hindbrain. The placode rapidly invaginates into the underlying mesenchyme to form the otic cup. Subsequently, the cup pinches off from the surface ectoderm and closes into the otic vesicle (or otocyst), which then

undergoes a drastic remodelling to give rise to the inner ear (reviewed in *Alsina and Whitfield, 2017*; *Basch et al., 2016a*).

How are the different cell types and territories of the inner ear specified and positioned during its embryonic development? The prevalent view is that diffusible signals (e.g. Wnt, Hedgehog, FGF, retinoic acid) secreted by tissues surrounding the otocyst direct the expression of key factors (transcription factors and signalling molecules) to specify various otic cell identities (reviewed in *Fekete and Wu, 2002*; *Ohta and Schoenwolf, 2018*). It was also proposed that once specified, adjacent but distinct populations of otic progenitors might be prevented from intermixing by lineage-restricted tissue boundaries akin to those discovered at the interface of 'embryonic compartments' in the *Drosophila* wing disc (*Brigande et al., 2000b*; *Fekete, 1996*). This 'compartment boundary' model has received some support from the analysis of mouse mutants exhibiting defects in restricted domains of the inner ear and some fate map studies, but it remains hypothetical with respect to the existence and precise location of tissue boundaries (*Kil and Collazo, 2002*).

One aspect of inner ear development that results in the spatial separation of adjacent groups of cells is the formation of its sensory organs. In the chicken or mouse otic vesicle, the prospective sensory domains, or prosensory domains, are recognisable as thickened epithelial patches in the ventro-medial and posterior regions (*Knowlton, 1967*). The prosensory patches express the Notch ligand Jag1, which promotes the maintenance of the prosensory fate by Notch-mediated lateral induction (reviewed in *Daudet and Żak, 2020*; *Kiernan, 2013*). They also express the high-mobility group transcription factor Sox2, which is essential for the formation of all inner ear sensory organs (*Dabdoub et al., 2008*; *Kiernan et al., 2005*), as well as otic neurogenesis (*Steevens et al., 2019*; *Steevens et al., 2017*). At early developmental stages, Sox2 marks a wide antero-posterior 'pansensory' domain occupying the ventro-medial wall of the otocyst (*Steevens et al., 2019*), due at least in part to the inhibition of Sox2 expression by Wnt activity in the dorsal aspect of the otocyst (*Żak and Daudet, 2021*). Subsequently, the pan-sensory domain splits into the posterior crista and several sensory organs in the anterior part of the otocyst: the anterior crista (AC), the lateral crista (LC), and the utricle (*Mann et al., 2017*; *Sánchez-Guardado et al., 2013*). This segregation process was first described in the inner ear of fish and amphibians more than a century ago (*Norris, 1892*) and could have played an essential role in the evolution of the vertebrate inner ear, by allowing new sensory organs to acquire specific functions (reviewed in *Fritzsch et al., 2002*). Genetic studies in the mouse have shown that mutations or absence of specific transcription factors and signalling molecules can lead to defects in sensory organ segregation (reviewed in *Alsina and Whitfield, 2017*). One such factor is Lmx1a (cLmx1b in the chicken), a LIM-homeodomain transcription factor whose orthologue Apterous acts as a 'selector gene' for the dorsal compartment of the *Drosophila* wing disc (*Rincón-Limas et al., 1999*). In the mouse inner ear, Lmx1a is expressed in the non-sensory territories separating the sensory organs, and its absence causes severe morphogenesis defects and partial fusions between adjacent sensory territories (*Koo et al., 2009*; *Nichols et al., 2008*; *Steffes et al., 2012*). Our previous work has shown that Lmx1a antagonises Jag1/Notch signalling within some early sensory-competent cells, thereby leading those cells to adopt a non-sensory fate between adjacent sensory organs (reviewed in *Daudet and Żak, 2020*; *Mann et al., 2017*). Live-imaging studies in zebrafish embryos have also shown that a progressive reduction in Jag1b expression occurs within the cells located between segregating sensory organs, confirming the central role played by the regulation of Notch activity in this process (*Ma and Zhang, 2015*). Despite these insights into the genetic regulation of sensory organ segregation, the cellular basis of this process remains unclear. Is it solely dependent on changes in cell character regulated by Notch and Lmx1a, or does it involve other processes and if so, of which type?

To tackle this question, we examined the changes in cell morphology during the segregation of the LC and AC cristae from the pan-sensory domain in the embryonic chicken and mouse otocyst. We identified a population of prosensory cells with enlarged surfaces, which appear at the interface of segregating organs and gradually elongate and re-align along the border of the prospective cristae before down-regulating Sox2 expression. As segregation proceeds, these large cells constrict basally at the interface of Lmx1a-positive (on the cristae side) and -negative (on the pan-sensory domain side) cells, strongly suggesting the formation of a lineage-restricted tissue boundary. In Lmx1a-null mouse otocysts, the boundary domain is disrupted; the AC and the LC initiate their segregation from the utricle but they remain fused to one another, indicating that Lmx1a is required for the differentiation

of the non-sensory cells separating the two cristae. Finally, we show that the perturbation of ROCK-dependent actomyosin contractility, a common player in the establishment of lineage-restricted boundaries, leads to defects in the organisation of the boundary domain and abnormal sensory organ segregation.

Altogether, our results strongly suggest that a lineage-restricted tissue boundary forms at the interface of segregating sensory organs, providing support to the compartment-boundary model of inner ear development. They also uncover striking similarities in the genetic circuits and cellular mechanisms of tissue segregation in the inner ear and the *Drosophila* wing disc.

## Results

### A specialised boundary domain forms between segregating vestibular organs in the chicken otocyst

We focused in this study on the formation of the AC and LC, using whole-mount preparations of the otic vesicle immunostained for the prosensory marker Sox2. In the chicken inner ear, the AC is the first to segregate from the large anterior pan-sensory domain between stages HH23 and HH25 (E4-E5), followed by the LC between HH25 and HH27 (E4.5-E5.5) (n=10) (*Figure 1a–c*; see also *Mann et al., 2017*). When we examined at high magnification phalloidin-stained samples, we noticed that the cells located at the interface of the cristae and the pan-sensory domain had a larger apical surface than those of surrounding territories.

Focusing on the LC, which could be identified from the earliest stages of its formation, we distinguished three stages of segregation as a function of Sox2 expression and the extent of separation from the pan-sensory domain. At stage 1 (HH25–26), the prospective LC formed a budding that was not yet separated from the pan-sensory domain (n=4) (*Figure 1a and d–e"*). Seven to nine rows of Sox2-positive cells located at the interface of the future LC and the pan-sensory domain appeared slightly larger than those located on either side. At stage 2 (HH26–27; *Figure 1b and g–h"*), Sox2 started to be down-regulated within the interface domain separating the LC from the pan-sensory domain (n=4). Scattered Sox2-positive nuclei were also present at the edge of a second Sox2-positive lobe of the LC, which remained connected to the pan-sensory domain. Of note, the second lobe of the LC does not give rise to sensory cells at later developmental stages and, as discussed later, it is not present in the mouse inner ear. In some cases, the large cells appeared to align their apical borders along the edge of the crista, forming a multicellular actin-cable-like structure, suggesting a coordinated increase in cell bond tension (*Figure 1g–h"*). At stage 3 (HH27–28; *Figure 1c and j–k"*), the two lobes of the LC were clearly separated from the pan-sensory domain by an 80- to 180-µm-wide territory containing cells with no (or very reduced) Sox2 expression (n=4). At the border of the LC, 7–10 rows of large cells remained visible. Comparable changes were observed at the border of the AC (*Figure 1—figure supplement 1*), although we were not able to analyse the early stages of segregation for this organ.

Concomitant to these changes in cell surface morphology, we noticed a progressive enrichment in F-actin staining at the base of the epithelium along the presumptive border of the LC (n=4) (*Figure 1f–f", i–i", and l–l"* and *Figure 2*). At stages 2–3, it was associated with a clear basal constriction of the cells and an upward displacement of the basal lamina along the presumptive border of the LC (*Figure 1l–l"* and *Figure 2b–c*). This suggests that the basal constriction could at least partly explain the enlarged apical surface of the cells composing the specialised boundary domain.

We next used the Epitool (*Heller et al., 2016*) and Icy software (see Materials and methods) to analyse in greater detail the changes in surface cell morphology during the segregation of the LC. Cell skeletons were generated from phalloidin-stained preparations (n=3 samples per stage), allowing the quantification of cell surfaces, elongation (the ratio between their long and short axis), and orientation (*Figure 3*). This analysis showed that the large cells at the interface of the LC and pan-sensory domain expanded their apical surface and increased in number from stage 1 to stage 3 (*Figure 3b–c*); they became more elongated than neighbouring cells from stage 2 (*Figure 3d*) and their long axis became gradually aligned along the same planar polarity vector, parallel to the edge of the crista, at stages 2–3 (*Figure 3e–f*). The increase in cell surface area was maximum in the interface domain but observed throughout the tissue, explaining the overall reduction in cell density observed between stage 1 and stage 3. The elongation and re-orientation of the cells were also noticeable throughout

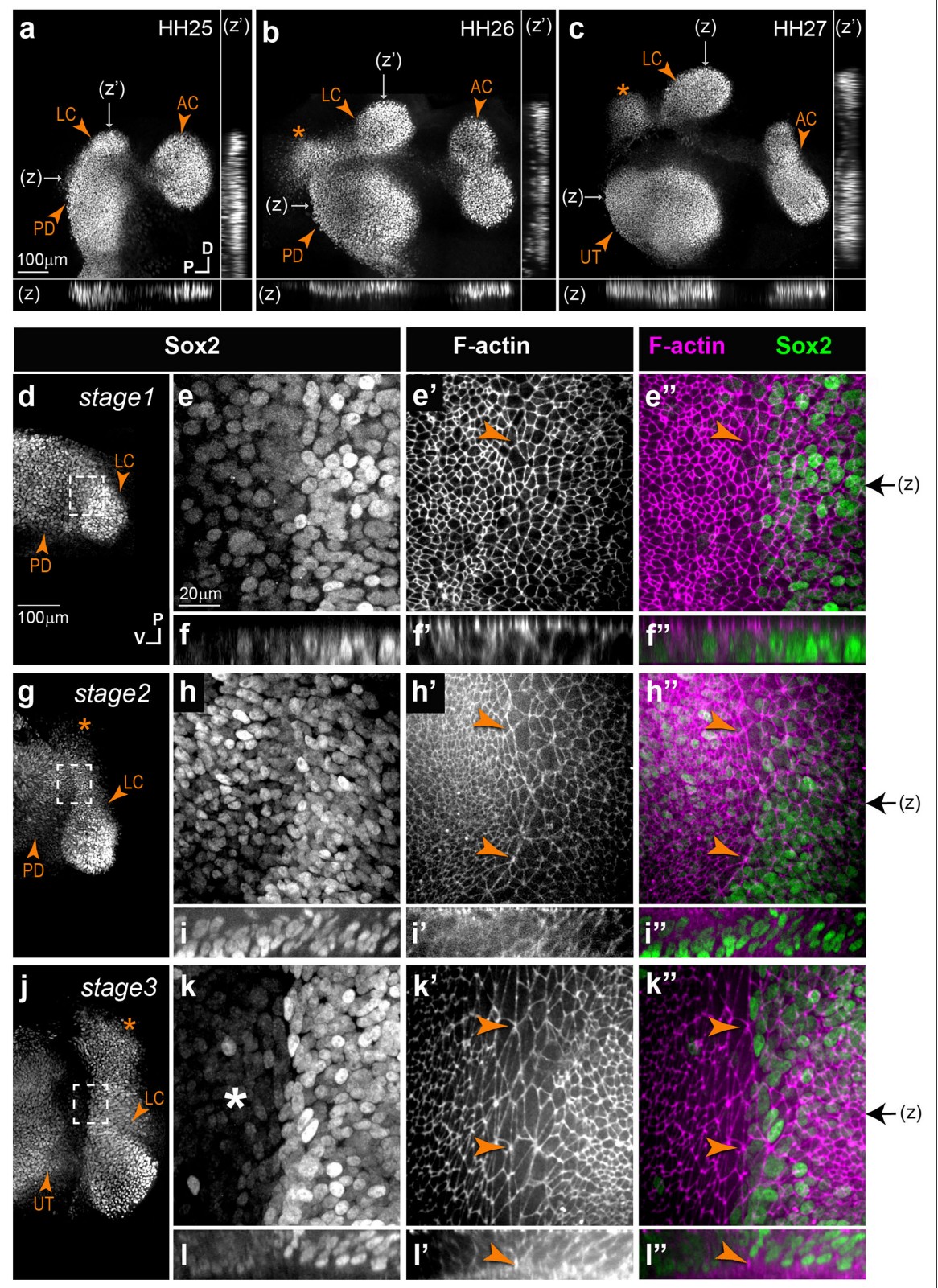

**Figure 1.** Segregation of the anterior (AC) and lateral cristae (LC) in the chicken inner ear. (**a–c**) Low-magnification surface views and transverse line optical reconstructions (**z,z'**) of whole-mount preparations of the anterior part of the chicken otocyst at different stages of development (HH = Hamburger and Hamilton stages), immunostained for Sox2 expression. The AC and LC segregate sequentially from the pan-sensory domain (PD), which then becomes the utricle (UT). The LC is connected to a second Sox2-expressing domain (stars) that does not give rise to a sensory organ. (**d–l"**) Stages

*Figure 1 continued on next page*

*Figure 1 continued*

1 (**d–f"**), 2 (**g–i"**), and 3 (**j–l"**) of the segregation of the LC, characterised by a progressive down-regulation of Sox2 expression in the domain separating the LC from the PD domain. High-magnification views (**e–e"**, **h–h"**, and **k–k"** correspond to the boxed areas in d–g–j, respectively) reveal striking changes in cellular morphology and organisation during LC segregation. A specialised boundary domain composed of large cells that progressively align their membranes along the border of the LC (arrowheads in **e'–l"**) becomes clearly visible from stage 2 onwards. This is concomitant to the formation of a basal constriction at the interface of the crista and pan-sensory domain, clearly visible at stage 3 in orthogonal z-reconstructions of the boundary domain (arrowheads in **l–l"**). Black arrows on the side of images **e"**, **h"**, and k" indicate the position of z-reconstructions.

The online version of this article includes the following figure supplement(s) for figure 1:

**Figure supplement 1.** Segregation of the anterior crista (AC) in the chicken inner ear.

the epithelium, suggesting that these processes are primarily driven by global anisotropic changes in tissue tension. As seen in LC samples stained for the apical surface marker ZO-1 and DAPI (*Figure 3— figure supplement 1a-a"*), the majority of cells with a large surface area were not undergoing mitosis, and their nuclei had a comparable size to those of neighbouring prosensory and non-sensory cells. Altogether, these results indicate that very dynamic changes in cell morphology and tissue patterning occur at the interface of the prospective LC and the pan-sensory domain before (stage 1) and during (stages 2–3) their segregation.

## A specialised boundary domain forms at the interface of Lmx1a-positive and Lmx1a-negative cells in the mouse otocyst

To find out if a comparable interface domain forms in the mammalian inner ear, we analysed the otocysts of *Lmx1a^{GFP}* knock-in mouse embryos. In a previous study, we showed that the inner ear of heterozygous *Lmx1a^{GFP/+}* mice develops normally, and GFP expression becomes gradually upregulated in the non-sensory territories surrounding the vestibular prosensory patches (*Mann et al., 2017*).

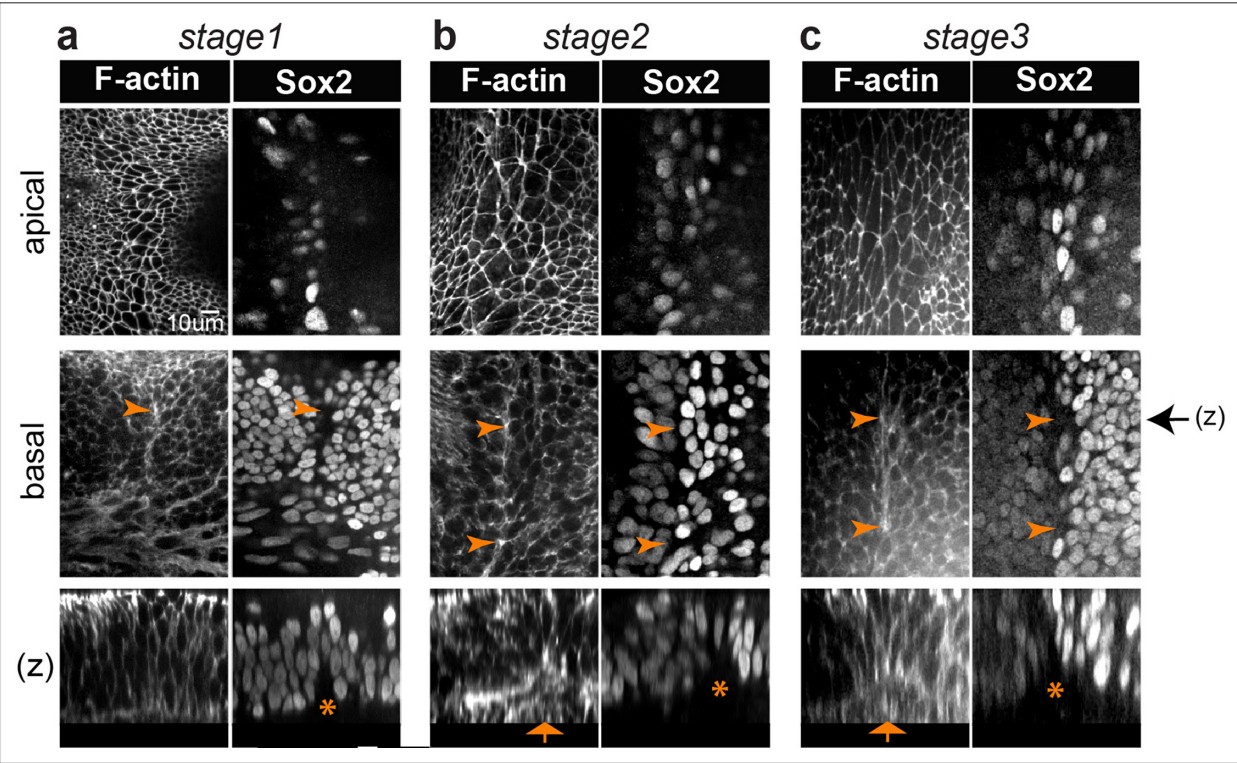

**Figure 2.** A multicellular basal constriction forms at the interface of the lateral crista (LC) and pan-sensory domain. (**a–c**) Samples stained for F-actin and Sox2 expression at stages 1–3 of LC segregation in the chicken inner ear. Optical sections collected from the surface and basal planes, and orthogonal reconstructions (**z**). In all images, the pan-sensory domain is on the left and the LC on the right. (**a**) F-actin is enriched at the base of the interface of the LC from stage 1 (arrowheads). (**b–c**) At stages 2–3, the F-actin enrichment becomes very clear along the prospective border of the LC (arrowheads). A basal constriction of the cells (arrows in z) and an upward displacement of the basal lamina (stars) are also noticeable. Black arrow on the side of images of the basal plane indicates the corresponding positions of z-reconstructions.

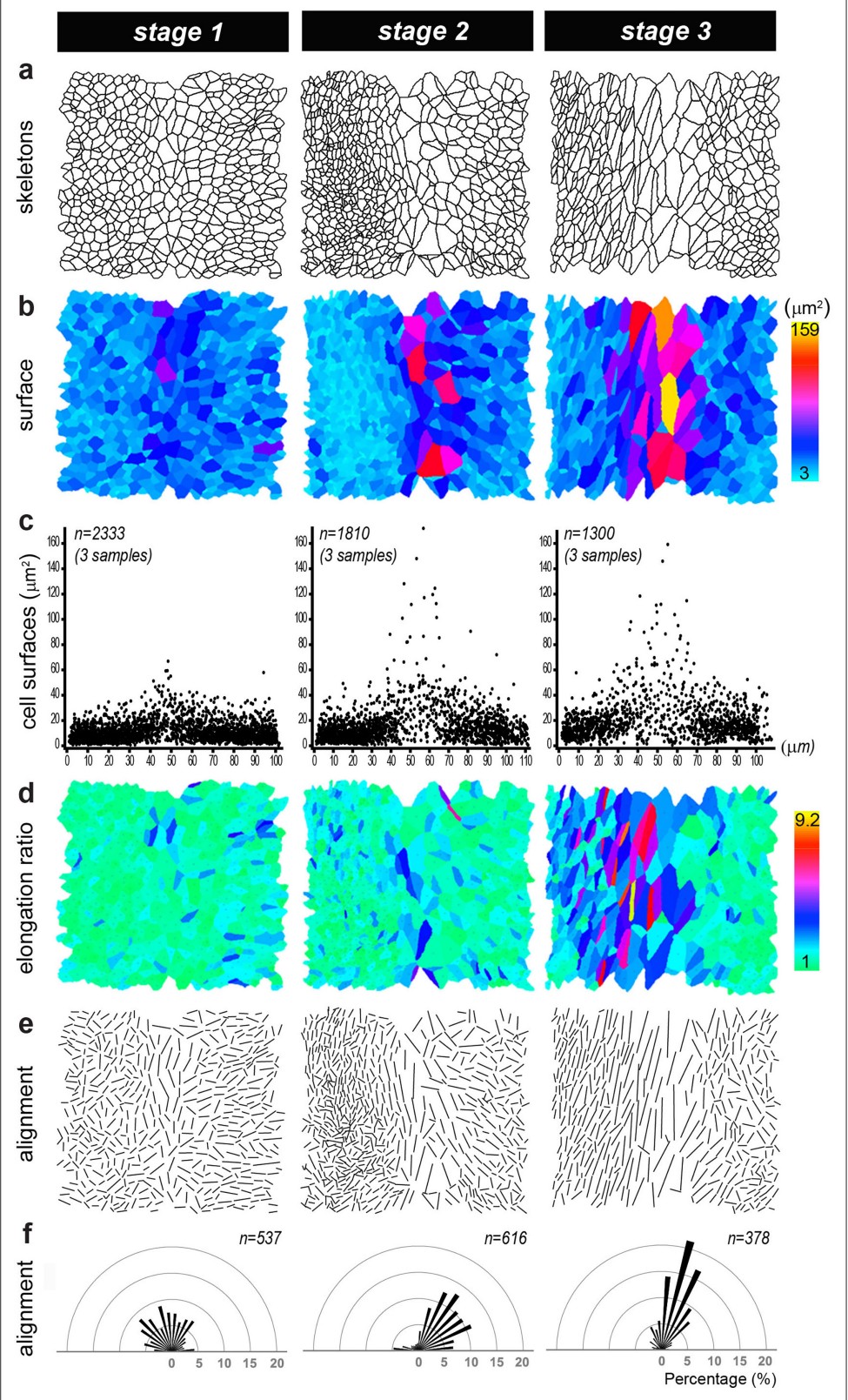

**Figure 3.** Analysis of the morphological changes during segregation of the lateral crista (LC) in the chicken otocyst. (**a**) Cell skeletons generated from phalloidin-stained preparations at three stages of LC segregation. In all images, the interface domain is in the middle, the pan-sensory domain on the left, and the LC on the right. (**b–c**) Colour scale and pooled scatter plots of the surface cell area show that the largest cells are located at the

*Figure 3 continued on next page*

*Figure 3 continued*

interface of the LC and pan-sensory domain from stage 1. (**d**) Colour-coded representation of the cell elongation ratio. (**e**) Visualisation of the long axis of the cells showing a progressive elongation between stages 1 and 3 and a clear alignment of the cells along the border of the LC at stage 3. (**f**) Angular distribution of cell orientations in the boundary domain shown in e relative to the longitudinal axis of the boundary domain. 'n' represents the number of cells analysed for each sample.

The online version of this article includes the following figure supplement(s) for figure 3:

**Figure supplement 1.** Visualisation of cell nuclei at the boundaries of segregating sensory organs.

However, we did not investigate in detail the earliest stages of segregation of the two cristae from the pan-sensory domain.

In E11.5 otocysts (n=4), two small Sox2-positive patches corresponding to the prospective AC and LC were visible at the anterior edge of the pan-sensory domain (*Figure 4a*). However, in contrast to what was observed in the chicken otocyst, the LC did not have a second Sox2-positive lobe, and the segregation of the AC and LC was synchronous. In fact, the AC was still connected to the pan-sensory domain at the time the LC budding emerged from it. In some way, the size and location of the AC and LC in E11.5 mouse otocysts resembled that of the two Sox2-positive lobes of the LC in the chicken otic vesicle at stage 2 (compare with *Figure 1b*). At this stage, Lmx1a-GFP expression pattern overlapped with that of Sox2 in between the presumptive cristae and at their interface with the large pan-sensory domain (*Figure 4b–b′*). Seen from the surface, the border of the Lmx1a-GFP-positive territory was relatively jagged, although GFP-positive and -negative cells remained clearly separated. The interface domain contained cells with a slightly enlarged surface in one out of four samples only. In two out of four E11.5 samples with good quality phalloidin staining, we observed segments of basal F-actin enrichment at the edge of the GFP-positive domain (*Figure 4c–c′* and *Figure 4—figure supplement 1a*). We conclude that E11.5 corresponds to an early stage of segregation of the cristae, equivalent to 'stage 1' in the chicken inner ear.

At E13.5 (n=5), Sox2 expression marked three discrete patches corresponding to the AC, the LC, and the prospective utricle (*Figure 4d*). Lmx1a-GFP expression was still present in Sox2-positive cells at the border of the AC and LC and in prospective non-sensory cells in between the two cristae which exhibited a clear reduction in Sox2 expression (*Figure 4e–e′*). Furthermore, an interface domain with enlarged cells was clearly recognisable in between the cristae and prospective utricle (*Figure 4f–f′* and *Figure 4—figure supplement 1b*). Remarkably, Lmx1a-GFP was detected in the half of the large cell domain facing the cristae; the other half, facing the prospective utricle, had no or very low levels of GFP fluorescence and exhibited reduced Sox2 expression. The apical membranes of the cells at the interface of GFP-positive and -negative domains were in some samples markedly aligned (*Figure 4—figure supplement 2*), suggesting an increase in cell bond tension along the border of Lmx1a expression.

In the basal planes, F-actin staining was enriched in multicellular cable-like structures, and a basal constriction similar to that observed in the chicken otocyst (at stages 2–3) was occasionally present at the precise interface of the Lmx1a-expressing and non-expressing cells (orthogonal reconstructions in *Figure 4f–f′* and *Figure 4—figure supplement 1b*).

In neonate mice (n=3; *Figure 4g*), Lmx1a-GFP expression persisted in a wide non-sensory territory separating the two cristae from the utricle. In contrast to earlier stages, there was no overlap between Sox2 and GFP expression at the border of the sensory organs (*Figure 4h–h′*), and a basal constriction was either absent or impossible to precisely locate in the samples examined (*Figure 4i–i′*).

Altogether, these data indicate that the cellular correlates of cristae segregation are comparable in the mouse and chicken inner ear, despite some differences in its dynamics and outcomes. In both species, a specialised population of boundary cells with enlarged apical surfaces appears at the interface of the two cristae and the utricle at early stages of their segregation. In the apical plane of the epithelium, cells elongate and align their borders in the middle of the large cell domain, at the precise interface of Lmx1a-expressing and non-expressing cells. In the basal planes, the interface of Lmx1a-GFP and positive cells is marked by the formation of multicellular actin-cable-like structures, and in some cases a basal constriction and upward displacement of the basal lamina. This strongly suggests that a lineage-restricted tissue boundary separates these two cell populations.

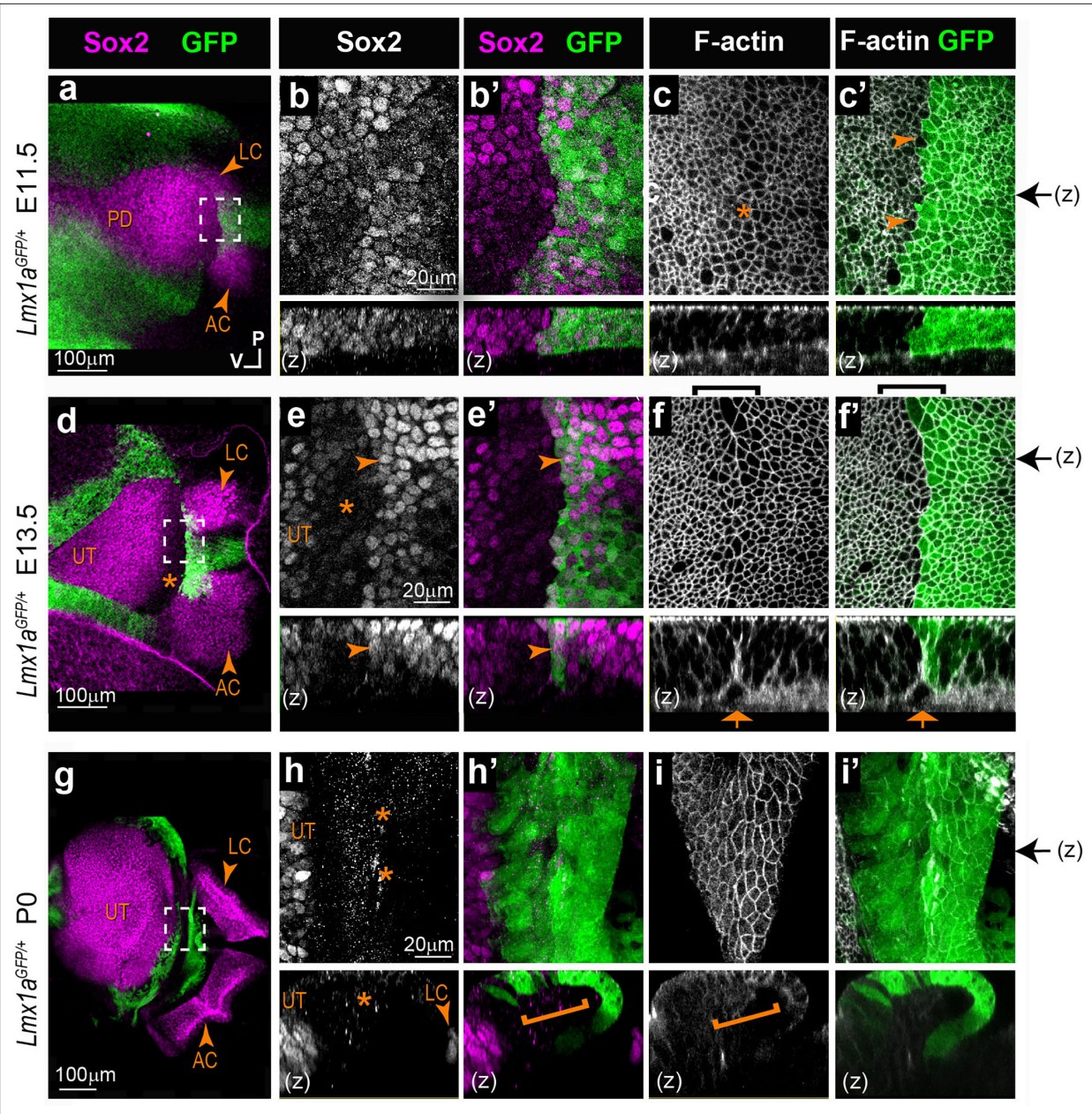

**Figure 4.** Segregation of the anterior (AC) and lateral cristae (LC) from the utricle in the heterozygous *Lmx1a* ^GFP/+^ knock-in mouse inner ear. Whole-mount surface views and orthogonal reconstructions (**z**) of the anterior part of the otocysts from *Lmx1a* ^GFP/+^ mouse at different stages of development, immunostained for Sox2 expression and stained for F-actin. (**a**) At E11.5, the prospective AC and LC are visible at the anterior edge of the pan-sensory domain (PD). Lmx1a-GFP expression pattern overlaps with that of Sox2 in between the presumptive cristae and at their interface with the PD (**b–b'**). The border of the Lmx1a-GFP positive territory is jagged, but GFP-positive and -negative cells are clearly separated (arrowheads in c'). Cell surfaces appear slightly enlarged at the limit of the Lmx1a-GFP (star in c). (**d**) At E13.5, the AC, LC, and UT are separated. Lmx1a-GFP expression is present in Sox2-positive cells at the border of the AC and LC (arrowheads in e–e') and in prospective non-sensory cells in between the two cristae, which exhibited a clear reduction in Sox2 expression. Cells with enlarged surface areas are clearly recognisable in between the LC and UT (brackets in f–f'). Lmx1a-GFP is detected in the half of the large cell domain facing the cristae. A basal constriction is present at the precise interface of the Lmx1a-expressing and non-expressing cells (arrows in f–f'). At P0, Lmx1a-GFP expression persisted in a wide non-sensory territory separating the two cristae from the UT (**g**). There is no overlap between Sox2 and GFP expression at the border of the sensory organs (stars and brackets in h–h'). No basal constriction is observed (bracket in i). Black arrows on the side of images c', f', and i' indicate the position of z-reconstructions.

The online version of this article includes the following figure supplement(s) for figure 4:

**Figure supplement 1.** Formation of a multicellular actin-cable-like structure during segregation of the anterior and lateral cristae from the utricle in *Lmx1a* ^GFP/+^ knock-in mouse.

**Figure supplement 2.** Alignment of cell borders at the interface of Lmx1a-positive and -negative cells in the mouse inner ear.

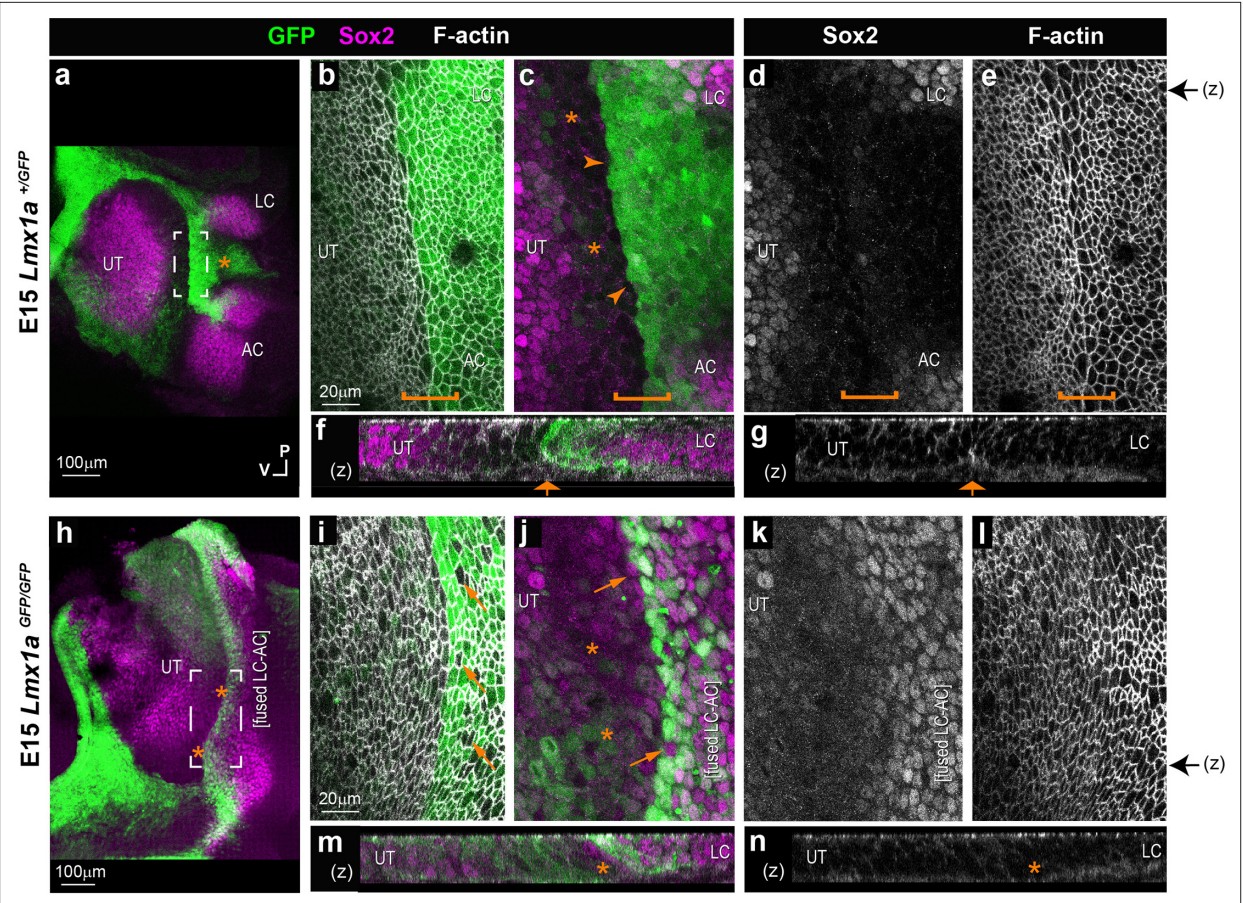

**Figure 5.** Comparison of the boundary domain in *Lmx1a* ^GFP/+^ and *Lmx1a* ^GFP/GFP^ mouse inner ear. Whole-mount preparations of the utricle (UT), lateral (LC), and anterior (AC) cristae region, after immunostaining for Sox2 (magenta) and F-actin labelling with phalloidin (white). (**a**) In the E15 *Lmx1a* ^GFP/+^ inner ear (**a–g**), the three sensory patches are segregated and Lmx1a-GFP is present around the utricle and in between the two cristae (star in a). High-magnification surface (**b–e**) and transverse reconstructions (**f–g**) of the interface domain (brackets). The boundary of Lmx1a-GFP expression is relatively straight (arrowheads), and Sox2 expression is reduced in the cells immediately adjacent to it and facing the utricle (stars in c). A basal constriction is visible at the border of the Lmx1a-GFP expression domain (arrows in f–g). In the E15 *Lmx1a* ^GFP/GFP^ inner ear (**h–n**), GFP expression is reduced at the interface of the UT and the fused AC and LC (stars in h). At higher magnification (**i–n**), note the presence of GFP-negative cells in between GFP-positive cells at the interface domain (arrows in i–l), but a domain of reduced Sox2 expression (stars in j) is maintained between the fused cristae and the UT. (**m–n**) In orthogonal reconstructions, note the absence of a basal constriction at the border of the cristae domain. Black arrows on the side of images e and l indicate the corresponding position of z-reconstructions in f–g and m–n, respectively.

The online version of this article includes the following figure supplement(s) for figure 5:

**Figure supplement 1.** Quantitative analysis of the boundary domain in *Lmx1a* ^GFP/+^ (left column) and *Lmx1a* ^GFP/GFP^ (right column) E15 mouse inner ear.

## The boundary domain is disrupted in *Lmx1a*-null mice

The complete absence of Lmx1a function leads to severe defects in inner ear morphogenesis, including the absence of semicircular canals or tissue constrictions and an abnormal growth and segregation of sensory organs (*Koo et al., 2009*; *Mann et al., 2017*; *Nichols et al., 2008*; *Steffes et al., 2012*). To find out if Lmx1a is required for the formation of the boundary domain, we compared cell morphology and organisation in E14 *Lmx1a* ^GFP/+^ heterozygous (n=8) versus *Lmx1a* ^GFP/GFP^ homozygous (*Lmx1a*-null; n=5) knock-in mice (*Figure 5*). Otocysts from homozygous *Lmx1a* ^GFP/GFP^ animals had very severe defects in the morphology of the AC and LC, which in all samples exhibited various degrees of expansion and fusion with one another (compare *Figure 5a and h*). Consistent with our previous observations (*Mann et al., 2017*), Sox2 expression was maintained in the GFP-positive cells within the fused cristae, suggesting that the fusion of the two cristae is, to a large extent, due to a specification defect: in the absence of Lmx1a, sensory-competent cells fail to differentiate into non-sensory cells at the interface of the AC and LC, producing instead a large and continuous sensory territory.

However, a relatively straight border of GFP expression was present at the edge of the fused cristae (*Figure 5b and i*), and a domain with reduced Sox2 expression was present between the cristae and utricle of the *Lmx1a^{GFP/GFP}* samples, although not as clearly defined as in the *Lmx1a^{GFP/+}* samples (stars in *Figure 5c–d and j–k*). This suggests that the initial segregation of the crista and utricle pools of sensory progenitors is, to some extent, independent from Lmx1a function. Nevertheless, there were some differences in GFP pattern, cell morphology, and organisation at the interface of the fused cristae and utricle. Firstly, the GFP pattern was very mosaic across *Lmx1a^{GFP/GFP}* cells, including at the border of the GFP-positive domain. This may be due to the abnormal differentiation of cells of the Lmx1a lineage, as well as increased mixing between cells belonging or not to the Lmx1a lineage at the edge of the cristae. Secondly, there was no basal constriction at the interface of the GFP-positive and -negative cells in the *Lmx1a^{GFP/GFP}* samples (compare *Figure 5f–g and m–n*). Finally, the cellular organisation of the boundary domain was markedly different in the *Lmx1a^{GFP/GFP}* samples. In fact, a quantitative analysis of cell surface morphology showed that cells on both sides of the boundary domain in *Lmx1a^{GFP/GFP}* samples (n=3) tended to be larger, more aligned with one another and more elongated than in *Lmx1a^{GFP/+}* ones (n=2) (*Figure 5—figure supplement 1*). Interestingly, these changes affected both GFP-positive and -negative domains, suggesting non-cell-autonomous consequences to the absence of *Lmx1a*.

Altogether, these results show that Lmx1a is required for the differentiation of the non-sensory territories between the AC and LC and the maintenance of the boundary domain between the cristae and the utricle. This is consistent with the notion that Lmx1a acts as a 'selector' for the non-sensory fate in the domain separating the two cristae. Lmx1a function is also required for the formation of the large cell domain and its associated basal constriction at the limit of the Lmx1a-expressing domain, but neither appears necessary for the initial separation of the utricle from the cristae.

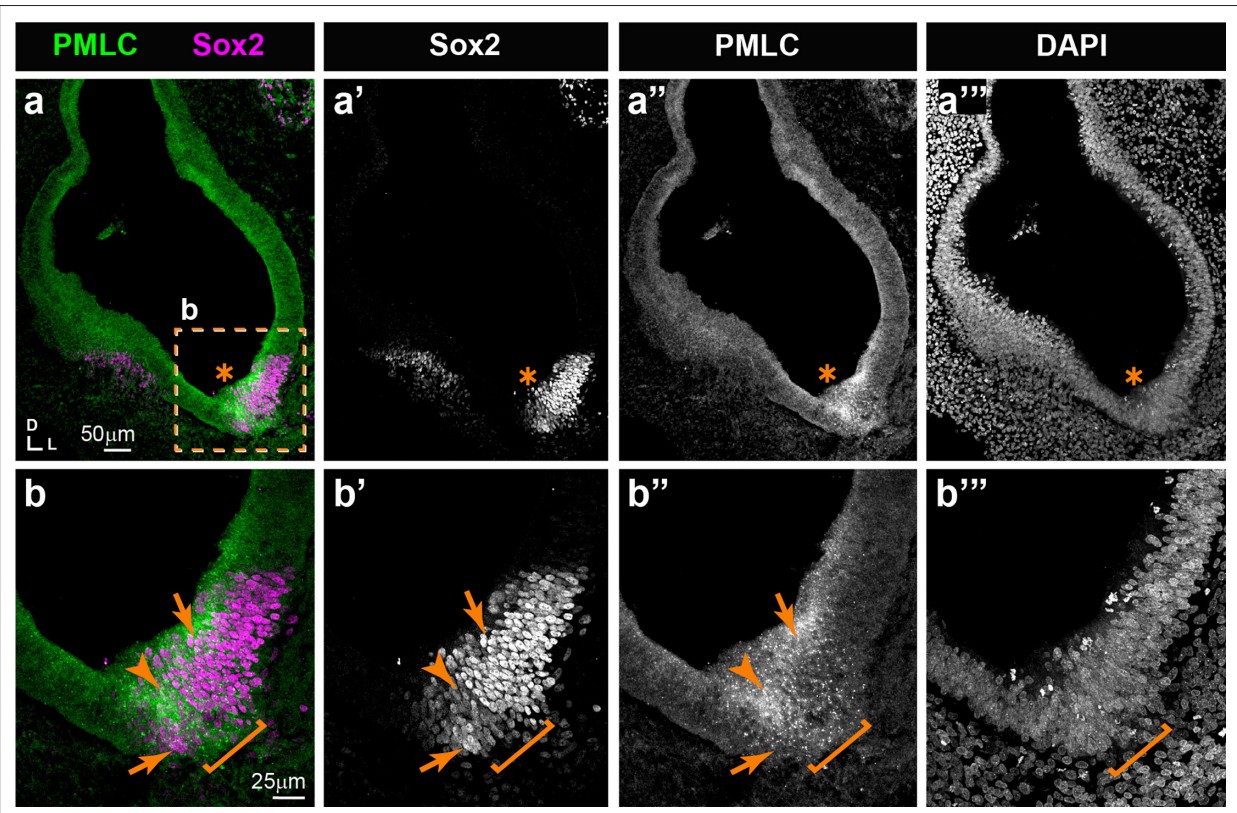

**Figure 6.** Expression pattern of phospho-myosin light chain (PMLC) in cross-sections of an E5 chicken inner ear. (**a–a'''**) Punctate PMLC staining is present in the ventral part of the inner ear where a Sox2-positive prosensory domain (star) is localised. (**b–b'''**) High-magnification views of the boxed area in a. The interface between the strong and weak Sox2 staining domains (arrows), typically corresponding to the boundary domain (bracket), exhibits a noticeable enrichment of PMLC staining at the surface and basal aspects of the epithelium (arrowheads).

# Pharmacological blockade of ROCK-dependent actomyosin contractility disrupts sensory organ segregation

In other contexts, tissue boundary formation or maintenance depends on increased actomyosin contractility at the interface of lineage-restricted compartments (reviewed in *Dahmann et al., 2011*; *Monier et al., 2011*). The resulting increase in cell bond tension is thought to create a physical 'fence' that restricts cell intermixing. To find out if this might also be the case in the chicken inner ear, we investigated the pattern of actomyosin activity using immunostaining for anti-phospho-myosin light chain 2 (PMLC) (*Figure 6*). The staining for PMLC was in general relatively weak and diffuse throughout

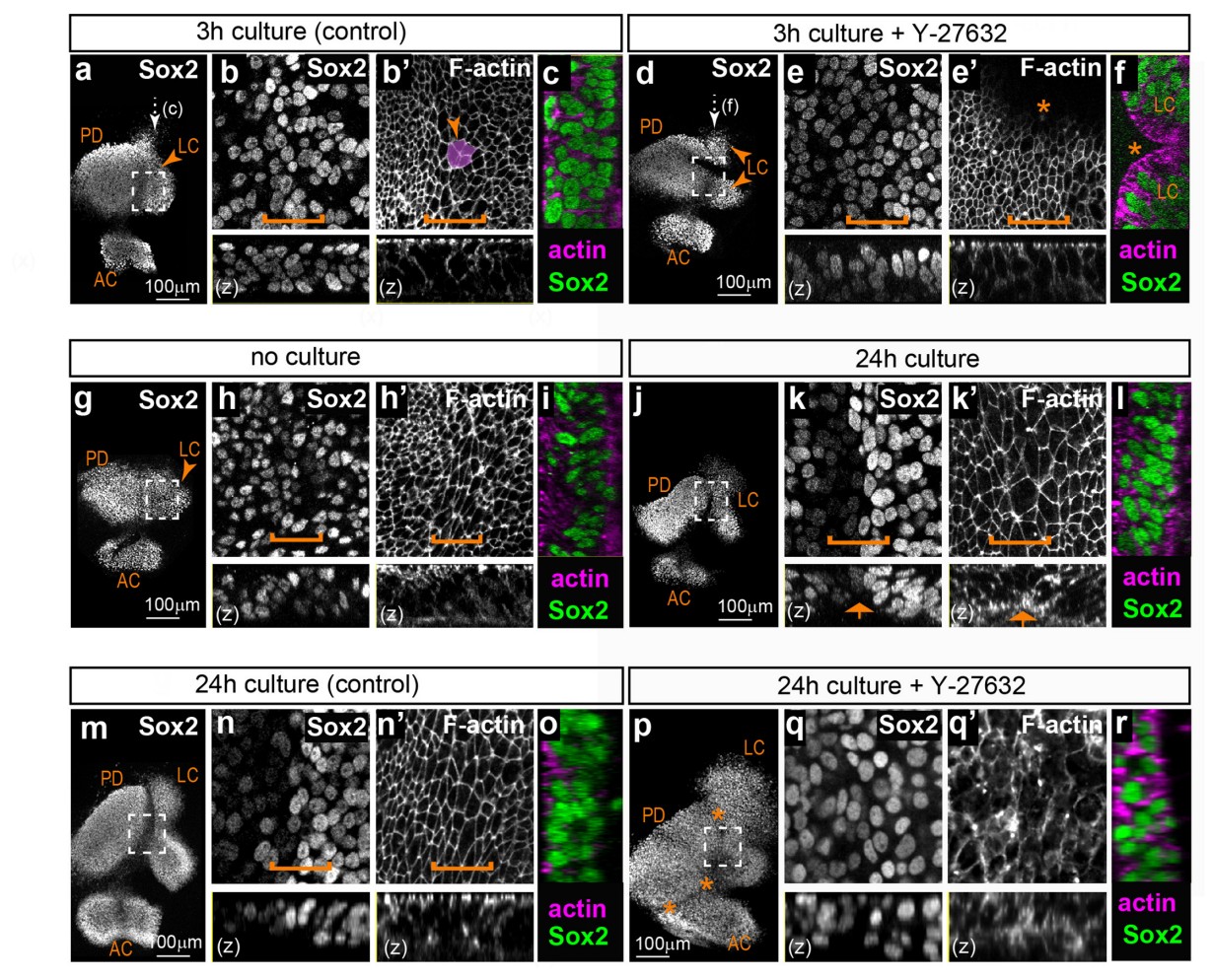

**Figure 7.** Effects of the ROCK inhibitor Y-27632 on the segregation of the lateral crista in organotypic cultures of embryonic chicken otocysts. (**a–f**) Surface and orthogonal views (**z, c, f**) of organotypic cultures of stage HH26 otocysts maintained for 3 hr in control medium (**a–c**) or medium supplemented with 20 µm Y-27632 (**d–f**). Panels b–c and e–f are high-magnification views of the boxed area in a and d, respectively; the position of the large cell domain is indicated by a bracket and an example of multicellular rosette is highlighted (arrowhead in b'). Treatment with Y-27632 induces a loss of multicellular rosettes and abnormal folding of the epithelium (stars in e'–f). (**g–l**) Surface and orthogonal views (**z, i, l**) of a pair of otocysts from the same HH26 embryo; one otocyst was fixed immediately (**g–i**) and the other maintained for 24 hr in culture (**j–l**). Panels h–i and k–l are high-magnification views of the boxed area in g and j, respectively; the position of the large cell domain is indicated by a bracket. After 24 hr, the lateral crista has progressed in its segregation from the pan-sensory domain. There is a clear reduction in Sox2 expression levels at the edge of the prospective LC and a basal constriction is noticeable (arrows in k–k'). (**m–r**) Surface and orthogonal views (**z, o, r**) of a pair of otocysts from the same HH26 embryo; one otocyst was cultured 24 hr in control medium (**m–o**) and the other maintained for 24 hr in 20 µm Y-27632 (**p–r**). Panels n–o and q–r are high-magnification views of the boxed area in m and p, respectively. Treatment with Y-27632 induces a loss of the typical epithelial cell morphology and large cell domain (compare n' and q') and a disruption of sensory organ segregation (stars in p). Abbreviations: LC = lateral crista; AC = anterior crista; PD = pan-sensory domain.

The online version of this article includes the following figure supplement(s) for figure 7:

**Figure supplement 1.** Quantification of the effects of ROCK inhibition on cell morphology and organisation during segregation of the lateral crista.

epithelial cells, but was markedly increased and punctate within Sox2-expressing prosensory domains. The boundary between segregating sensory organs also exhibited strong PMLC punctate staining extending from the surface to the base of the epithelium (n=7). We then set up free-floating organo-typic cultures of HH25 chicken otocysts, an early stage of segregation of the LC, and treated these with Y-27632, a potent inhibitor of ROCK-dependent actomyosin contractility (*Figure 7*).

After short-term (3 hr) culture in control DMEM medium (n=3; *Figure 7a–c*), HH26 otocysts at early stages of LC segregation had a well-recognisable large cell domain, but only one out of three samples examined had a clear basal constriction. Multicellular rosettes were frequently observed at the surface of the epithelium (*Figure 7b'*). However, the interface between the Sox2-positive nuclei of the LC and utricle was not as well defined as in freshly fixed tissue, suggesting a partial disruption of tissue organisation in vitro. The boundary domain was still present in samples treated for 3 hr with 20 µM Y-27632 (n=3; *Figure 7d–f*), but there was no basal constriction. Multicellular rosettes were completely absent and we noted in all samples some folding and invaginations of the epithelium, possibly caused by the inhibition of cortical actomyosin contractility by Y-27632. Nevertheless, the interface between the LC and the utricle was still visible, indicating that a short blockade of actomyosin contractility does not result in extensive cell mixing at the interface of the two organs. Furthermore, a quantitative analysis of the effects of Y-27632 treatment (*Figure 7—figure supplement 1a–f*) did not show significant changes in the size and alignment of surface cells in treated samples compared to controls.

Next, we examined the consequences of longer Y-27632 treatments. We first performed some experiments to determine whether the initial stages of LC segregation can occur ex vivo. We compared LC size and segregation in 15 pairs of otocysts, with each pair from the same embryo (HH26 – stage 1 or 2 of LC segregation) composed of an otocyst immediately fixed after dissection and the other one maintained in culture for 24 hr before fixation (*Figure 7g–l*). All of the cultured LC showed some progression in LC formation compared to their contralateral control, with a clear boundary between weakly and strongly Sox2-expressing cells in 12 out of 15 samples and a basal constriction present in 14 out of 15 samples. The size of the LC was greater in cultured samples, suggesting growth and survival of the prosensory cells. However, the distance between the LC and the pan-sensory domain ranged from 15.82 µm to 49.82 µm only, which was reduced compared to what would normally be observed in ovo in a stage HH27 otocyst (n=9; distance = 46.3–179.62 µm; see also *Mann et al., 2017*). These results show that the initial stages of specification and segregation of the LC do occur in vitro, albeit at a slower pace than in ovo.

We next investigated cell morphology and organisation in contralateral HH26 otocysts cultured for 24 hr in either control medium or 20 µm Y-27631 (*Figure 7m–r*). Only three out of 6 Y-27631-treated samples could be analysed, due to the extreme fragility of the tissue which made the dissection and mounting of the samples impossible. Compared to their contralateral controls, the Y-27631-treated samples exhibited an extensive fusion and expansion of both the AC and LC domains, which were nevertheless present and distinguishable from the pan-sensory domain. Signs of severe epithelial disorganisation included the frequent absence of F-actin staining at the apical surface, suggesting a disassembly of the adherens junction complexes, and Sox2-positive nuclei located very close to the surface. The interface domain did not exhibit the typical down-regulation in Sox2 expression (*Figure 7p–q*) and had a statistically significant reduction in the number of large cells compared to controls (*Figure 7—figure supplement 1a–d*).

These results indicate that ROCK activity might be dispensable for the specification and growth of the LC, but it is absolutely required for the maintenance of its epithelial character and its segregation from the pan-sensory domain.

## Perturbation of ROCK activity in ovo disrupts the organisation of the boundary domain

In light of the very severe defects in epithelial morphology caused by long-term Y-27631 treatment and the limited extent of cristae segregation in vitro, we turned to a genetic approach to interfere with ROCK-dependent actomyosin contractility in ovo in the developing otocyst. We used in ovo electroporation (EP) of the otic cup to overexpress in the otocyst a truncated dominant-negative form of ROCK fused to GFP (RIIC1-GFP), able to bind to its upstream regulator Shroom3, but lacking the kinase domain necessary for Myosin II regulation (*Nishimura and Takeichi, 2008*). Pilot experiments on HH23 samples transfected with a transient expression vector confirmed that RIIC1-GFP

localises at the cell membrane (*Figure 8a*). The analysis of surface cell morphologies with Epitool (*Figure 8b–d*) showed that RIIC1-GFP causes a mild (approximately 23%) but statistically significant (p<0.01, Mann-Whitney test U=10622.5) increase in the average apical surface area of transfected otic cells (18.22 µm²; SD = 11.04) compared to untransfected cells (14.84 µm²; SD = 8.38), consistent with an inhibition of cortical actomyosin contractility. In comparison, cells transfected with a control RCAS-GFP construct (*Figure 8e*) do not have significantly enlarged apical surface area (p=0.56, Mann-Whitney test U=77504). Next, we used the RCAS retroviral system (*Hughes, 2004*) to investigate the effects of RIIC1-GFP on cristae segregation, since EP of a replication-competent retrovirus proviral DNA leads to a sustained and widespread transgene expression (*Freeman et al., 2012*). The right otic cups of E2 (HH13–14) embryos were electroporated with either the RCAS-GFP (control) or the RCAS-RIIC1-GFP proviral DNA, and the embryos were incubated for 3–5 days post-EP. In controls (n=8; *Figure 8g–h'''*), the LC was present in six out of eight samples and separated from the utricle by distances ranging from 55 µm to 155 µm. We hypothesise that the absence of LC in two of the controls may be due to abnormal or delayed development of the inner ear after EP. The pattern of GFP expression was very variable among samples, but in general, it encompassed sensory and non-sensory regions. However, in two samples, we noted a very marked difference in GFP fluorescence levels on each side of the boundary of the LC (*Figure 8h'*). Although several factors could potentially explain this observation (e.g. different rate of GFP production or degradation in different cell populations), it fits with the idea that there is limited cell intermixing at this particular location.

In the RCAS-RIIC1-GFP electroporated samples (n=6; *Figure 8i–l'''*), the LC was detected in all cases but it was either connected to the utricle (n=2/6) or migrated at a smaller distance than in controls (n=4/6; *Figure 8f*), suggesting an impaired segregation. The examination of the putative LC boundary domain at high magnification showed that as previously observed with the short-term overexpression experiments, RIIC1-GFP fluorescence was enriched at the cell surface and baso-lateral membranes (*Figure 8j' and l'*).

In four out of six analysed samples, clear defects in the alignment of Sox2-positive cells were observed at the border of the prospective LC in regions with high levels of RCII-GFP expression, suggesting potential intermixing between cells of the LC and those of prospective non-sensory domains. Although it was not always possible to visualise it clearly, a basal constriction appeared to be present, including in domains with high levels of RCII-GFP expression (*Figure 8j'–j'''*). Strikingly, in one sample with a mosaic pattern of RCII-GFP expression in the boundary domain, the untransfected cells were the only ones exhibiting a clear alignment of their apical borders reminiscent of what was observed in controls (*Figure 8l'–l'''*).

We conclude that ROCK-dependent actomyosin contractility is required for the coordination of cortical cell tension at the boundary between the LC and pan-sensory domain and for normal sensory organ segregation. However, additional mechanisms are likely to limit the intermixing of prosensory cells at the boundary since the separation of Sox2-high and Sox2-low cells was still visible in these experiments.

## The boundary domain is highly proliferative

One important function of actomyosin-dependent tissue boundaries is to limit the disruptive impact of cell proliferation on the spatial organisation of cells during tissue growth and morphogenesis (reviewed in *Monier et al., 2010*). This prompted us to examine the spatial pattern of cell proliferation during LC segregation. To detect S-phase cells, we administered in ovo a short (2 hr) pulse of the thymidine analogue EdU to stage HH23–25 embryos before collecting inner ear tissue and processing it for Click-iT EdU reaction and Sox2 immunostaining. Numerous EdU-positive cells were present throughout the boundary domain at both early (stage 1, *Figure 9a–b*; n=7) and advanced (stages 2–3, *Figure 9c–d*; n=7) segregation stages, and these always included some of the cells located at the interface of the prospective LC and pan-sensory domain. This was confirmed by the seemingly random distribution of mitotic figures (visualised by DAPI staining) within the boundary domain of fixed samples (n=7 for each stage; *Figure 9b, d, and e*). We conclude that the boundary at the interface of the LC and pan-sensory domains contains mitotic cells throughout the segregation stages.

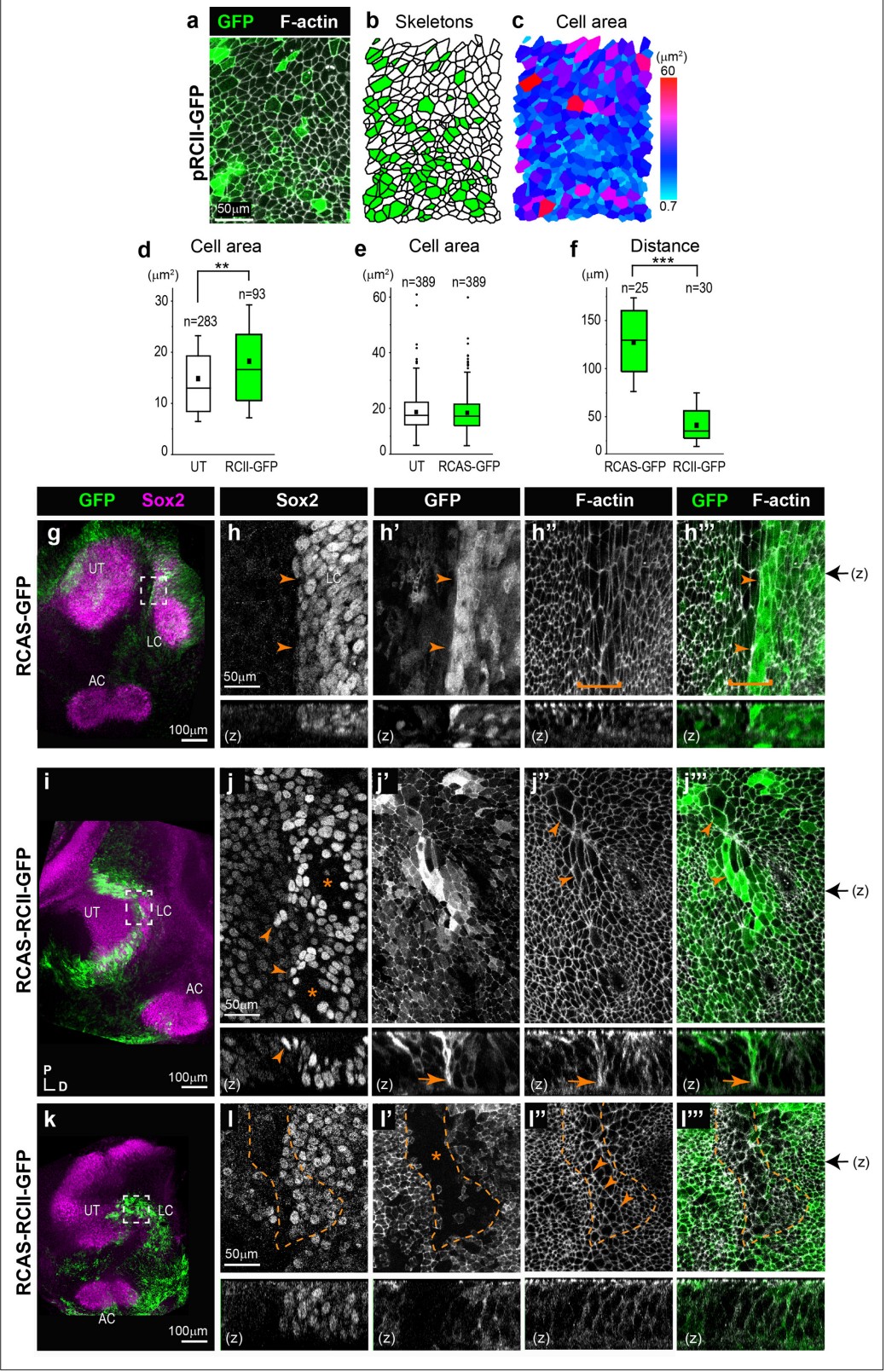

**Figure 8.** Overexpression of a dominant-negative form of ROCK in the developing chicken inner ear disrupts sensory organ segregation. (**a**) Surface view of otic epithelial cells 24 hr after in ovo electroporation with the pRCII-GFP expression plasmid. (**b–e**) Analysis of cell skeletons of transfected (in green) versus untransfected (utricle [UT]) cells indicates that overexpression of the RCII-GFP fusion protein induces an increase in apical cell surface area

*Figure 8 continued on next page*

*Figure 8 continued*

compared to control electroporation with RCAS-GFP (**e**). (**f**) Quantification of the distance between the lateral crista (LC) and UT in ears transfected with pRCII-GFP expression plasmid and control RCAS-GFP. (**g–h'''**) Surface views of an E6 control otocyst electroporated at E2 with an RCAS-GFP proviral DNA. In this sample, the limit of Sox2 expression (arrowheads in h) matches that of strong GFP fluorescence (arrowheads in h' and h'") and the presumptive tissue boundary in the middle of the large cell domain (bracket in h" and h'") separating the LC from the UT. (**i–l'''**) Surface views of two E6 samples electroporated at E2 with RCAS-RCII-GFP and showing abnormal segregation of the LC. In the first example, the interface domain contains poorly aligned Sox2-expressing cells (arrowheads in j). A basal constriction is still visible (arrows in j'–j''', transverse reconstructions), but the large cells of the interface domain are not properly aligned (arrowheads in j" and j'"). In the second example (**k–l'''**), a mosaic pattern of RCII-GFP fluorescence is observed in the interface domain. Cells with low levels of RCII-GFP fluorescence (star in l') exhibit a more regular alignment of their apical borders (arrowheads in l") than transfected cells. The black arrows on the right side of images h'", j'", and l'" indicate the corresponding position of z-reconstructions.

## Discussion

We discovered a specialised actomyosin-dependent boundary domain at the interface of segregating sensory organs in the developing inner ear. We propose that this boundary restricts intermixing between Lmx1a-positive and -negative cells during the segregation of adjacent sensory organs and acts as a signalling centre guiding the differentiation of non-sensory territories. Our findings provide strong support to the 'compartment boundary' model for inner ear morphogenesis (*Brigande et al., 2000b*; *Fekete, 1996*; *Fekete and Wu, 2002*) and suggest that an evolutionary conserved module for tissue boundary formation has been co-opted in the inner ear for separating its distinct sensory organs.

### A tissue boundary forms during the segregation of inner ear sensory organs

The morphogenesis of complex tissues depends on the harmonious coordination of cell proliferation, cell specification, and tissue patterning events. Tissue boundaries play an important role in this process by preventing, at specific locations, the intermingling of cells destined to form distinct anatomical structures. Besides this 'physical' role, tissue boundaries can act as signalling centres or 'organisers', regulating the proliferation and differentiation of adjacent cells through various intercellular signalling pathways. Tissue boundaries were discovered (*Garcia-Bellido et al., 1973*) and most studied in the imaginal wing disc of *Drosophila*, where they establish lineage-restricted compartments along the dorso-ventral and antero-posterior axis (*Wang and Dahmann, 2020*). In vertebrates, they are present in the nervous system (*Addison and Wilkinson, 2016*; *Langenberg and Brand, 2005*), in the ectoderm of the limb bud (*Altabef et al., 1997*), or in the gut (*Smith and Tabin, 2000*). Their formation typically involves three consecutive steps (*Addison and Wilkinson, 2016*; *Dahmann et al., 2011*; *Wang and Dahmann, 2020*). Firstly, 'selector' genes (encoding transcription factors) assign a specific identity to different populations of cells. Secondly, these cells interact through signalling molecules and sort themselves into adjacent domains according to their genetic identity through adhesive or repulsive interactions. Finally, actomyosin-dependent processes increase cell tension at the interface of adjacent compartments to create and maintain a stable 'fence' preventing cell mixing (*Major and Irvine, 2006*; *Monier et al., 2011*; *Monier et al., 2010*).

The morphogenesis of the inner ear has long been proposed to depend on the formation of lineage-restricted embryonic compartments analogous to those present in the fly wing disc. In fact, some genes have sharply defined domains of expression in the otic vesicle before any sign of morphological differentiation, and their absence prevents the formation of specific structures of the adult inner ear, suggesting that these may act as selector genes for a particular otic fate (*Brigande et al., 2000b*; *Fekete, 1996*; *Fekete and Wu, 2002*). Fate map and lineage studies have also suggested the existence of lineage-restricted boundaries in the dorsal part of the chicken otic vesicle, which gives rise to the endolymphatic duct and sac, although their exact location and cellular features remain unknown (*Brigande et al., 2000a*; *Kil and Collazo, 2002*; *Sánchez-Guardado et al., 2014*). In this study, we provide strong evidence that the segregation of the AC and LC from the prospective utricle is associated with the formation of a tissue boundary. It is composed of a subset of prosensory

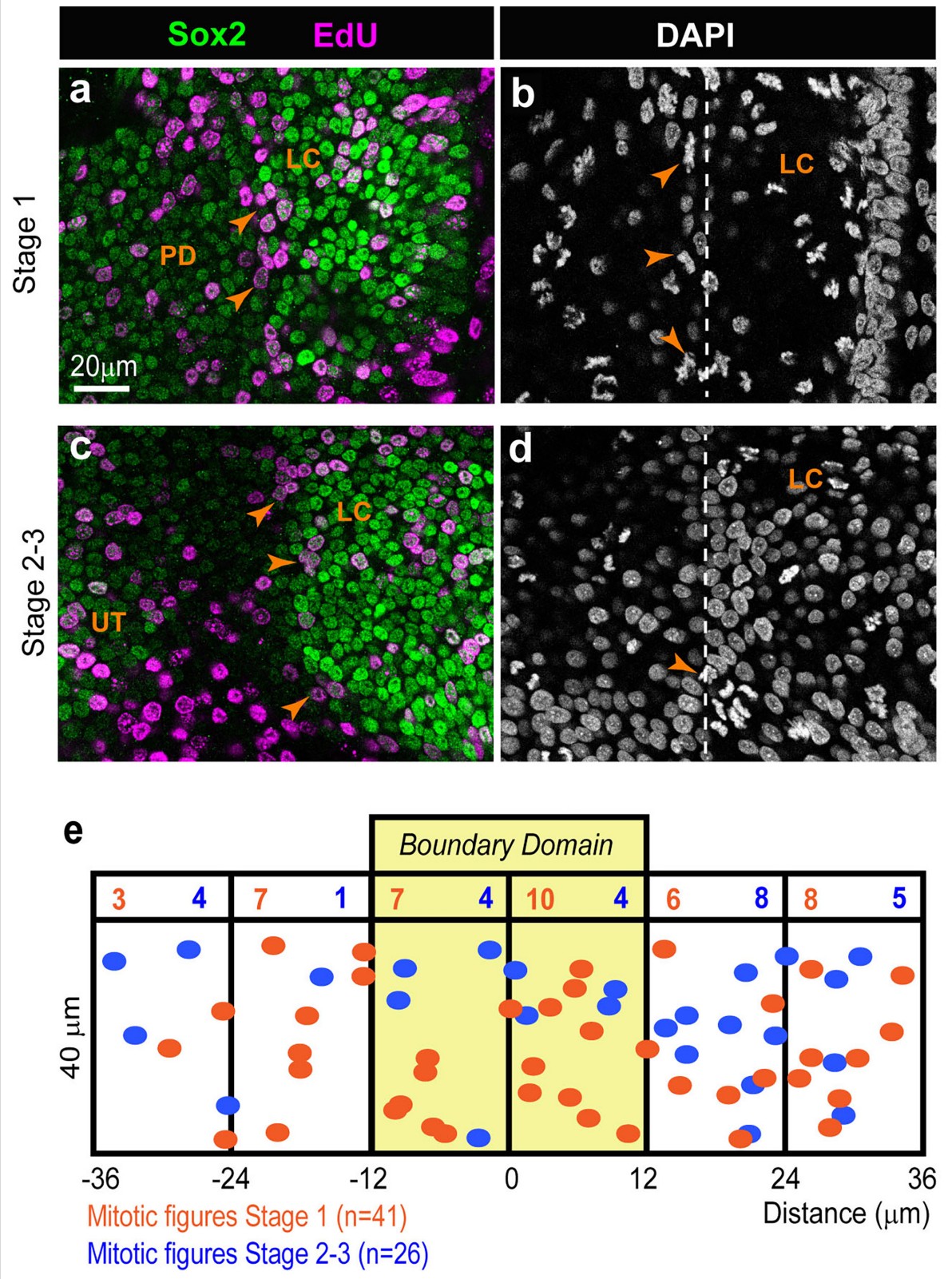

**Figure 9.** Analysis of cell proliferation during the segregation of the lateral crista (LC) in the chicken otocyst. Whole-mount preparations of samples collected 1 hr after an EdU pulse in ovo at stage 1 (**a–b**) or stages 2–3 (**c–d**) of segregation of the LC, immunostained for EdU, Sox2, and labelled with DAPI. Numerous EdU-positive cells (arrows in a–c) and mitotic figures (arrows in b–d) are present at the interface between the LC and PD (dotted line in b–d) at both stages. (**e**) Spatial distribution of the mitotic figures within a 36×72 µm² region centred on the middle of the boundary domain, for early and late stages of segregation (n=7 embryos for each stage).

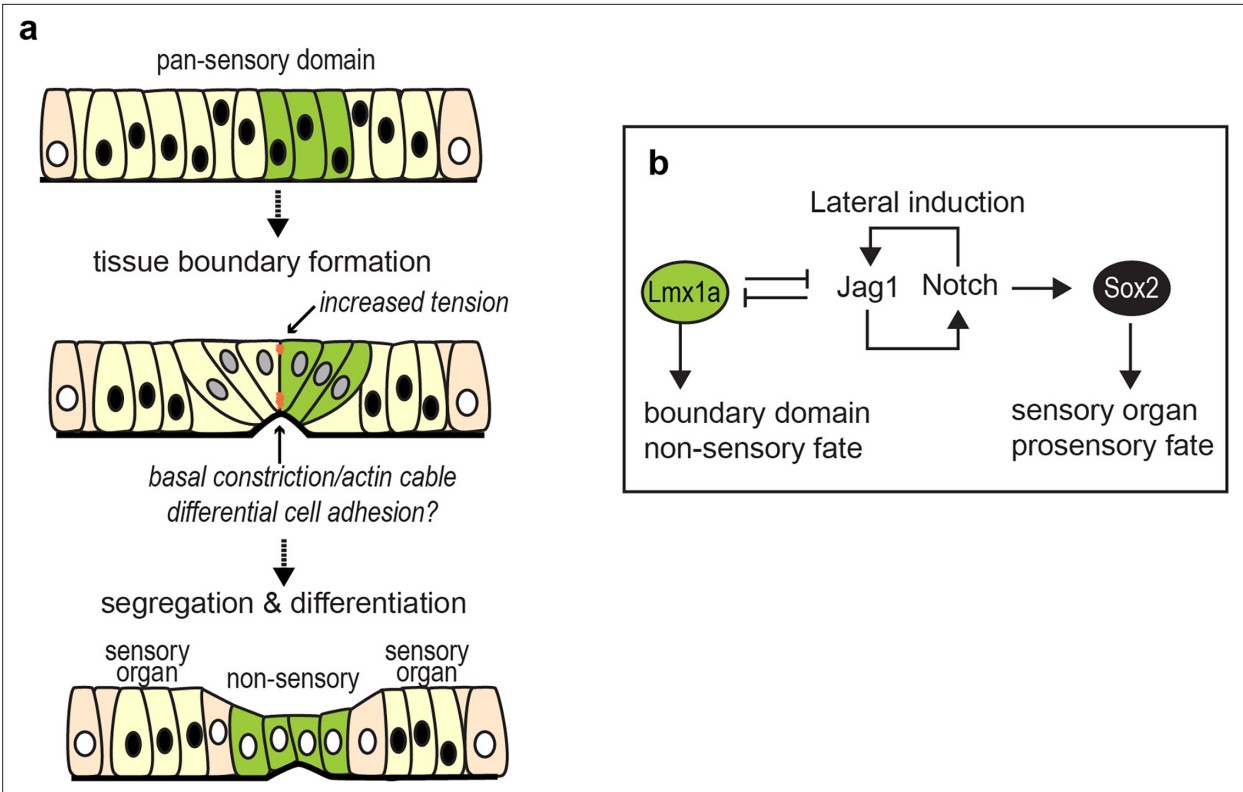

**Figure 10.** A schematic representation of sensory organ segregation. (**a**) The up-regulation of Lmx1a expression within the pan-sensory domain coincides with the formation of a specialised boundary domain composed of cells with enlarged cell surfaces. At the interface of Lmx1a-expressing and non-expressing cells, cells align their borders apically and form multicellular basal constrictions and actin-cable-like structures at the base of the epithelium. We propose that it is a transient lineage-restricted boundary, dependent on actomyosin contractility, which separates adjacent pools of sensory progenitors. As the spatial segregation proceeds, the Lmx1a-expressing cells give rise to non-sensory cells separating sensory organs. (**b**) Hypothetical regulatory gene network regulating sensory organ segregation. Notch-dependent lateral induction maintains Sox2 expression and promotes the prosensory fate. Lmx1a antagonises Notch activity and promotes adoption of a non-sensory fate; it is also required for the proper formation of the boundary domain between the cristae and utricle.

cells which progressively enlarge, elongate, and align their apical cell borders at the precise junction between Lmx1a-positive and negative cells (*Figure 10*). In the absence of lineage-tracing or live-imaging experiments, we do not have definitive proof that these cells generate a strict lineage-restricted boundary. However, their position (at the interface of two distinct 'genetic compartments') and cellular features (basal constriction, basal multicellular actin-cable-like structure, and alignment of their apical cell borders) are consistent with this idea. Furthermore, a lineage-restricted boundary could explain why we observed, in some of our RCAS-infection experiments, an uneven distribution of GFP-expressing cells on either side of the basal constriction separating the LC from the utricle. In a previous study, we argued that the changing patterns of Lmx1a and Sox2 expression in the embryonic inner ear suggested some form of dynamic competition for the adoption of the sensory versus non-sensory fates at the lateral border of sensory organs, as opposed to the existence of a strict lineage boundary (*Mann et al., 2017*). Our new results, which focused on earlier stages of sensory organ formation, call for a revision of this conclusion: the early prosensory cells are indeed labile, but their initial separation into distinct pools of sensory progenitors (for the cristae and the utricle at least) appears to be associated with the formation of a transient lineage-restricted tissue boundary.

Which mechanisms establish and maintain the boundary domain between segregating sensory organs? In a previous study, we showed that Lmx1a is essential for the specification of the non-sensory territories separating sensory organs and its overexpression in the chicken inner ear antagonises prosensory specification, suggesting that it acts as a 'selector' gene for the non-sensory fate (*Mann et al., 2017*). The present study confirms this hypothesis: in its absence (in *Lmx1a^{GFP/GFP}* embryos), cells of the Lmx1a lineage commit to a prosensory fate, leading to a fusion between the lateral and anterior

cristae. On the other hand, the fused cristae and the utricle are separated by a Sox2-negative domain, suggesting that the initial segregation of these two sensory domains proceeds independently from Lmx1a function. Nevertheless, the typical features (large cell domain and basal constriction) of the boundary domain separating the cristae from the utricle were absent in all but one of the *Lmx1a*-null samples examined. Altogether, these results suggest that Lmx1a plays a role in the maintenance or continued differentiation of the boundary domain, but is not required for its initial formation. Finally, the patterning abnormalities in *Lmx1a*^GFP/GFP^ samples occurred in both GFP-positive and negative territories, which points at some type of interaction between Lmx1a-expressing and non-expressing cells, and the possibility that the boundary domain is also a signalling centre influencing the differentiation of adjacent territories.

Given its conserved role in tissue boundaries, we hypothesised that actomyosin contractility could contribute to the formation or maintenance of the boundary domain. The in vitro results showed that short-term inhibition of ROCK-dependent actomyosin contractility with Y-27632 disrupts the morphology and patterning of boundary cells. Interestingly, the large cells were still present, but more elongated and more polarised than in control cultures. This suggests that global anisotropic forces within the developing otocyst contribute to a large extent to the elongated cellular shapes and long axis alignments observed at the interface of segregating organs. Nevertheless, the spatial separation of cells expressing either high or low levels of Sox2 was maintained. Longer-term treatments led to an apparent fusion of adjacent sensory territories, but this may have been caused by the complete loss of epithelial integrity observed in these experiments. On the other hand, the overexpression in the chicken inner ear of a dominant-negative form of ROCK, unable to interact with its upstream regulator Shroom3 (*Nishimura and Takeichi, 2008*), produced more informative results. Firstly, the LC failed to segregate normally from the pan-sensory domain, indicating a requirement for ROCK activity in this process. Secondly, the border of the LC was very irregular, with some intermingling between cells expressing high and low levels of Sox2. This would be expected if an actomyosin-dependent lineage boundary had been disrupted. Altogether, these results show that ROCK-dependent coordination of actomyosin contractility is crucial for the segregation of the sensory organs and the maintenance of a straight boundary between these, yet other mechanisms are likely to contribute to the initial sorting of the Lmx1a-expressing and non-expressing cells. One strong candidate is the basal constriction and actin-cable-like structure at the edge of the cristae, which resembles the one described at the midbrain-hindbrain boundary (*Gutzman et al., 2018*; *Gutzman et al., 2008*) and could contribute to the physical separation of sensory progenitors. It has been proposed that a zone of non-proliferating cells could impede cell movements between adjacent compartments at the dorso-ventral wing disc boundary (*Blair, 1993*; see however *O'Brochta and Bryant, 1985*, for a different view) or between rhombomeres in the hindbrain (*Guthrie et al., 1991*). However, the results of our EdU-incorporation assays show extensive proliferation of the cells composing the boundary domain at early and late stages of segregation, arguing against this hypothesis. Other mechanisms involved in boundary formation in the hindbrain and insect compartments include differential cell adhesion or repulsive interactions mediated by eph-ephrin signalling (*Dahmann et al., 2011*; *Wilkinson, 2018*). It is also possible that the repositioning of cells through medial intercalation could contribute to the straightening of the boundary, as well as the widening of the non-sensory territories in between sensory patches. Further studies will be needed to determine which of these mechanisms, if any, contributes to sensory organ segregation in the inner ear.

## Deep homologies between sensory organ segregation and *Drosophila* wing disc compartmentalisation: co-option of a tissue boundary module?

An increase in cortical cell tension or a multicellular actin cable has been described along many tissue boundaries (*Fagotto, 2014*; *Wang and Dahmann, 2020*): the dorso-ventral (*Major and Irvine, 2006*; *Major and Irvine, 2005*) and antero-posterior (*Landsberg et al., 2009*) boundaries of the wing disc, the parasegments (*Monier et al., 2010*) and salivary gland placode (*Röper, 2012*) boundaries in *Drosophila* and at the interface of rhombomeres of the hindbrain (*Calzolari et al., 2014*) in vertebrates. Hence, the implication of actomyosin-dependent processes to prevent cell intermixing is a conserved feature of tissue boundaries, which is also present in the inner ear. Perhaps more surprisingly, the core elements of the gene regulatory network regulating sensory organ segregation in the

inner ear and the dorso-ventral compartmentalisation of the *Drosophila* wing disc are remarkably similar (*Daudet and Żak, 2020*). In fact, Notch signalling acts by lateral induction in both tissues: to maintain prosensory character in the inner ear or to enhance Notch activity at the dorso-ventral boundary in the wing disc. We have argued that *Lmx1a* and *Apterous*, which are orthologous genes, play a selector-like function in both tissues, specifying the identity of dorsal cells in the wing disc and that of the non-sensory cells separating the vestibular organs in the inner ear. In both tissues, the interactions between Apterous/Lmx1a and Notch signalling have a comparable outcome: a reduction in the levels of Notch activity within Apterous/Lmx1a-expressing cells. In the wing disc, Apterous drives the expression of Fringe, a glycosyl-transferase that makes the Notch receptor less sensitive to the Serrate ligand in dorsal cells, potentiating signalling through Delta, expressed by ventral cells (*de Celis et al., 1996*; *de Celis and Bray, 1997*; *Doherty et al., 1996*; *Irvine and Wieschaus, 1994*; *Irvine and Wieschaus, 1994*; *Panin et al., 1997*; *Rauskolb et al., 1999*). We do not know yet if any of the vertebrate Fringe orthologues contribute to the regulation of Notch activity at the border of inner ear vestibular organs. However, it has been shown that Lfng and Mfng are expressed at the medial border of the prosensory domain in the embryonic mouse cochlea, and *Lnfg;Mfng* double mutants exhibit patterning defects consistent with an abnormal positioning of the medial border of the organ of Corti (*Basch et al., 2016b*). Further studies should investigate the roles of Fringe proteins during vestibular organ segregation. By extension, it would be worth investigating the potential role of Wnt signalling in this process, since one of the important consequences of Notch activity at the dorso-ventral wing disc boundary is to induce the expression of Wingless, which then regulates the growth and patterning of the wing (*de Celis et al., 1996*; *Diaz-Benjumea and Cohen, 1995*; *Rulifson and Blair, 1995*). It is worth noting that a similar genetic circuit including Notch signalling (Fringe, Jagged) and Lmx1 is deployed in vertebrate limbs to position the apical ectodermal ridge, the domain that acts as a growth zone and lineage boundary between the dorsal and the ventral ectoderm of the limb (*Altabef et al., 1997*). This 'deep homology', defined as 'the sharing of the genetic regulatory apparatus that is used to build morphologically or phylogenetically disparate animal features' (*Shubin et al., 2009*), suggests that Notch, Lmx1, and actomyosin contractility are part of an ancestral module for tissue boundary formation that has been co-opted and adapted for the patterning of a great variety of tissues.

Within the inner ear itself, it is possible that modifications of this core module are responsible for some of the interspecies differences in sensory organ numbers and morphologies. It is thought that the ancestral inner ear resembled that of jawless vertebrates (the lamprey and the hagfish), with only two cristae (anterior and posterior) and a single ventral macula (*Hammond and Whitfield, 2006*; *Higuchi et al., 2019*; *Jørgensen et al., 1998*; *Lowenstein et al., 1968*). Jawed vertebrates harbour three semicircular canals and cristae, as well as additional sensory organs in the vestibular and ventral part of the inner ear (*Retzius, 1884*; *Retzius, 1881*), enabling in particular the detection of air-borne sound in land vertebrates. Differences in inner ear sensory organ numbers, size, and morphologies are commonplace among vertebrates, and it is possible that some of these originate from changes in the genetic networks controlling sensory organ segregation. In the present study, we found at least a clear indication of interspecies variations in the segregation of the cristae: in the chicken inner ear, the anterior crista segregates before the lateral one, whilst in the mouse the segregation of the two cristae occurs simultaneously. Furthermore, the chicken LC is formed by two Sox2-positive lobes, one of which turns into a non-sensory territory. The fact that the specification of prosensory cells and their segregation is regulated by signalling pathways with well-known spatial patterning activities (FGF, Wnt, and Notch signalling) may have offered ample opportunities for 'sensory organ experimentation', facilitating in the course of vertebrate evolution the emergence of additional organs with new functionalities or, on the contrary, the suppression of redundant ones.

## Materials and methods
### Animals
Fertilised White Leghorn chicken (*Gallus gallus*) eggs were obtained from Henry Stewart UK and incubated at 37.8°C for the designated times. Embryonic stages refer to embryonic days (E), with E1 corresponding to 24 hr of incubation or to Hamburger and Hamilton stages (*Hamburger and Hamilton, 1951*). Embryos older than E5 were sacrificed by decapitation.

*Lmx1a^GFP* knock-in mice (*Griesel et al., 2011*) were maintained on a C57BL/6 mixed background at the UCL Ear Institute animal facility and bred to generate *Lmx1a^GFP/GFP* (homozygous/Lmx1a null) and *Lmx1a^GFP/+* (heterozygous) mice.

All animal procedures were carried out in accordance with the United Kingdom legislation outlined in the Animals (Scientific Procedures) Act 1986 and approved by the UK Home Office in the Project Licence PP8324029.

Both male and female animals were used in the study, and each animal served as an individual biological replicate. Each experiment was repeated at least three times, and within each experiment, a minimum of four samples was analysed.

## In ovo EP

EP of the otic placode/cup of E2 chick embryos (stage HH10–14) was performed using a BTX ECM 830 Electro Square Porator as previously described (*Freeman et al., 2012*). The RIIC2-GFP plasmid was obtained from *Nishimura and Takeichi, 2008*, and the insert was subcloned into RCAS using conventional methods. All plasmids used for in ovo EP were purified using the 16 QIAGEN Plasmid Plus Midi Kit (QIAGEN). The total concentration of DNA for each set of experiments ranged between 0.5 and 1 µg/µl. The electroporated inner ears were collected from the embryos between E5 and E7 and processed for whole-mount immunohistochemistry.

## Immunohistochemistry

Entire chicken embryos (E3–E4) or their heads (>E5) were collected, fixed for 1.5–2 hr in 0.1 M phosphate-buffered saline (PBS) with 4% paraformaldehyde and processed for whole-mount immunostaining as described previously (*Mann et al., 2017*). In brief, the inner ear epithelium was dissected out from surrounding mesenchyme and incubated for 30 min in a blocking solution consisting of PBS with 0.3% Triton and 10% goat serum. The primary and secondary antibodies (diluted in PBS with 0.1% Triton, or PBT) were incubated overnight at 4°C or for 2 hr at room temperature. Several rinses in PBT were done after antibody incubation. The following reagents were used: rabbit anti-Sox2 (Abcam, UK; 97959, 1:500); mouse IgG1 monoclonal anti-Sox2 (BD Biosciences, San Jose, CA, USA; 561469, 1:500); mouse IgG1 anti-phospho-Myosin light chain 2 (Ser19) (Cell Signaling Technology, MA, USA, 3675, 1:200), rat IgG1 anti-Laminin-β1 (Abcam, AB44941, 1:50); secondary goat anti-mouse IgG antibodies, and phalloidin conjugated to Alexa dyes (1:1000, Thermo Fisher Scientific UK). At the end of the immunostaining protocol, the samples were further dissected under a Leica MZ16F stereomicroscope equipped with a fluorescence light source to visualise precisely the position of the Sox2-expressing sensory organs and facilitate their mounting in Vectashield (Vector labs, UK). A fine layer of vacuum grease was applied between the slide and coverslip to avoid excessive flattening of the tissue. Confocal stacks were acquired using a Zeiss LSM880 inverted confocal microscope or a Perkin Elmer spinning-disc confocal and further processed with ImageJ.

## Analysis of cell morphologies

Phalloidin labelling was used to visualise and analyse apical cell surfaces. Confocal z-stacks of the surface of the samples were processed with ImageJ and exported into Epitool for automated cell segmentation (*Heller et al., 2016*). The cell skeletons were manually corrected using the CellEdit plug-in in the Icy software (*de Chaumont et al., 2012*). Colour scale of cell surface area and elongation ratio was generated based on the corrected cell skeleton using CellOverlay plug-in in Icy. Quantitative data on cell morphologies were exported into Excel spreadsheet, and scatter plots, radial histogram, box plots, and statistics of the data were generated using OriginPro 2017 and R. For the quantification of cell surfaces in Y-27632 experiments (*Figure 7—figure supplement 1*), a subset of cells located within a ± standard deviation interval from the location of the largest cells of each sample was analysed.

## Organotypic cultures

The anterior domain of the otocysts and its underlying mesenchyme were dissected from E5 (HH26) chicken embryos in room temperature L-15 medium (Leibovitz). The dissected tissues were either fixed at t=0 or cultured in 150–300 µl DMEM medium (Invitrogen) containing 1% HEPES and 20 µm Y-27632 (Sigma-Aldrich, cat # Y0503, lot # 103M4711V; dissolved in DMEM) in a tissue culture incubator (5%

$CO_2$, 37 degrees) for 3 hr or 24 hr. For the 3 hr-long experiments, the tissues were placed in 96-well plates and free-floating in DMEM medium. For the 24 hr-long experiments, the tissues were placed in 35 mm Mattek dishes and coated with a small (approximately 20 µl) drop of 0.4–0.8% agarose (low gelling temperature, Sigma-Aldrich) in 0.1 M PBS before adding DMEM medium. The cultured tissues were then fixed and processed for immunostaining as previously described.

## Cell proliferation assays

Fertilised eggs were windowed at E4–5 (HH23–25) and a 200 µl drop of a PBS solution containing 200 µM of EdU (5-ethynyl-2'-deoxyuridine) was applied on top of the embryo. The eggs were sealed with tape and returned to incubation for 2 hr, then the embryos' heads were collected and fixed for 30 min at room temperature in PBS with 4% formaldehyde. Following several rinses in PBS and PBT, the inner ear tissue was dissected and processed for Click-iT EdU reaction (Thermo Fisher UK) according to the manufacturer's protocol. At the end of the reaction, the tissue was further processed for Sox2 immunostaining as described above and labelled with DAPI before mounting. For the mapping of mitotic figures (identified by DAPI staining), a $72 \times 36$ µm$^2$ 'region-of-interest' centred on the boundary between the LC and pan-sensory domain was selected from a confocal stack acquired with a 40x objective. The position of each mitotic figure was recorded in x,y, and the results plotted for different samples collected at stage 1 (n=7) or 2–3 (n=7).

## Measurement of the distance between segregating sensory organs

The distance between the LC and the utricle was measured in maximum intensity projections of z-stacks using ImageJ. The distance (µm) was measured in five different positions along the boundary domain in each analysed sample. Next, all the results were analysed in R using two-sample Kolmogorov-Smirnov test and two-sided Mann-Whitney-Wilcoxon test.

## Acknowledgements

We thank Thea Støle and Caitlin Broadbent (UCL Ear Institute) for excellent technical support and Prof. Jonathan Gale for scientific discussions. This work was supported by an Action on Hearing Loss International Research Grant (G76 to ND and MŻ), a Royal National Institute for Deaf People Fellowship (P27 to MŻ) and the UK Research and Innovation (Medical Research Council MR/S003029/1 and MR/W005123/1 to ND and MŻ, Biotechnology and Biological Sciences Research Council BB/L003163/1 to ND, and for microscopy Biotechnology and Biological Sciences Research Council BB/R000549/1 to Jonathan Gale).

## Additional information

### Funding

| Funder | Grant reference number | Author |
| --- | --- | --- |
| Action on Hearing Loss | G76 | Magdalena Żak<br>Nicolas Daudet |
| Royal National Institute for Deaf People | P27 | Magdalena Żak |
| Medical Research Council | MR/S003029/1 | Magdalena Żak<br>Nicolas Daudet |
| Medical Research Council | MR/W005123/1 | Magdalena Żak<br>Nicolas Daudet |
| Biotechnology and Biological Sciences Research Council | BB/L003163/1 | Nicolas Daudet |
| Biotechnology and Biological Sciences Research Council | BB/R000549/1 | Nicolas Daudet |

| Funder | Grant reference number | Author |
|---|---|---|

The funders had no role in study design, data collection and interpretation, or the decision to submit the work for publication.

## Author contributions

Ziqi Chen, Formal analysis, Investigation, Visualization, Methodology, Writing – review and editing; Magdalena Żak, Data curation, Formal analysis, Supervision, Funding acquisition, Investigation, Visualization, Project administration, Writing – review and editing; Shuting Xu, Conceptualization, Formal analysis, Investigation, Visualization; Javier de Andrés, Formal analysis, Investigation, Visualization; Nicolas Daudet, Conceptualization, Formal analysis, Supervision, Funding acquisition, Investigation, Methodology, Writing - original draft, Project administration, Writing – review and editing

## Author ORCIDs

Magdalena Żak (iD) https://orcid.org/0000-0003-0888-6822

## Ethics

All animal procedures were carried out in accordance with United Kingdom legislation outlined in the Animals (Scientific Procedures) Act 1986 and approved by the UK Home Office in the Project Licence PP8324029. Both male and female animals were used in the study, and each animal served as an individual biological replicate. Each experiment was repeated at least three times, and within each experiment, a minimum of four samples were analysed.

Reviewer #1 (Public review): https://doi.org/10.7554/eLife.106305.3.sa1
Reviewer #3 (Public review): https://doi.org/10.7554/eLife.106305.3.sa2
Author response https://doi.org/10.7554/eLife.106305.3.sa3

---

# Additional files

## Supplementary files

MDAR checklist

## Data availability

Raw images and quantification data are available from UCL Research Data Repository (https://doi.org/10.5522/04/30383656.v1).

The following dataset was generated:

| Author(s) | Year | Dataset title | Dataset URL | Database and Identifier |
|---|---|---|---|---|
| Chen Z, Zak M, Xu S, de Andrés J, Daudet N | 2025 | A tissue boundary orchestrates the segregation of inner ear sensory organs | https://doi.org/10.5522/04/30383656.v1 | Figshare, 10.5522/04/30383656.v1 |

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
