## [Editor Report · eLife Assessment]

This is an **important** study describing the morphological changes during boundary formation between sensory and non-sensory tissues of the inner ear. The authors provided **solid** evidence that a transcription factor, Lmx1a and ROCK-dependent actinomyosin are key for border formation in the inner ear. However, future studies will be needed to investigate the direct relationships among boundary formation, Lmx1a and ROCK. This work will be of interest to developmental biologists interested in boundary formation.

---

## [Referee Report · Reviewer #1 (Public review)]

Summary:

This manuscript investigated the mechanism underlying boundary formation necessary for proper separation of vestibular sensory end organs. In both chick and mouse embryos, it was shown that a population of cells abutting the sensory (marked by high Sox2 expression) /nonsensory cell populations (marked by Lmx1a expression) undergo apical expansion, elongation, alignment and basal constriction to separate the lateral crista (LC) from the utricle. Using Lmx1a mouse mutant, organ cultures, pharmacological and viral-mediated Rock inhibition, it was demonstrated that the Lmx1a transcription factor and Rock-mediated actomyosin contractility is required for boundary formation and LC-utricle separation.

Strengths:

Overall, the morphometric analyses were done rigorously and revealed novel boundary cell behaviors. The requirement of Lmx1a and Rock activity in boundary formation was convincingly demonstrated.

Weaknesses:

However, the precise roles of Lmx1a and Rock in regulating cell behaviors during boundary formation were not clearly fleshed out. For example, phenotypic analysis of Lmx1a was rather cursory; it is unclear how Lmx1a, expressed in half of the boundary domain, control boundary cell behaviors and prevent cell mixing between Lmx1a+ and Lmx1a- compartments? Well-established mechanisms and molecules for boundary formation were not investigated (e.g. differential adhesion via cadherins, cell repulsion via ephrin-Eph signaling). Moreover, within the boundary domain, it is unclear whether apical multicellular rosettes and basal constrictions are drivers of boundary formation, as boundary can still form when these cell behaviors were inhibited. Involvement of other cell behaviors, such as directional cell intercalation and oriented cell division also warrant consideration. With these lingering questions, the mechanistic advance of the present study is modest.

Revision: The clarity of the text was improved. The open questions regarding the mechanisms were not experimentally addressed but discussed.

---

## [Referee Report · Reviewer #3 (Public review)]

Summary:

Lmx1a is an orthologue of apterous in flies, which is important for dorsal-ventral border formation in the wing disc. Previously, this research group has described the importance of the chicken Lmx1b in establishing the boundary between sensory and non-sensory domains in the chicken inner ear. Here, the authors described a series of cellular changes during border formation in the chicken inner ear, including alignment of cells at the apical border and concomitant constriction basally. The authors extended these observations to the mouse inner ear and showed that these morphological changes occurred at the border of Lmx1a positive and negative regions, and these changes failed to develop in Lmx1a mutants. Furthermore, the authors demonstrated that the ROCK-dependent actomyosin contractility is important for this border formation and blocking ROCK function affected epithelial basal constriction and border formation in both in vitro and in vivo systems.

Strengths:

The morphological changes described during border formation in the developing inner ear are interesting. Linking these changes to the function of Lmx1a and ROCK dependent actomyosin contractile function are provocative.

Weaknesses:

There are several outstanding issues that need to be clarified before one can pin the morphological changes observed being causal to border formation and that Lmx1a and ROCK are involved.

Comments on the latest version:

The revised manuscript has provided clarity of their results on some levels, but unfortunately, the basal restriction during border formation remains unclear and the study did not advance the understanding of role of Lmx1a in boundary formation. Overall comments are indicated below:

(1) The authors states in the rebuttal, "we do not think that ROCK activity is required for the formation or maintenance of the basal constriction at the interface of Lmx1a-expressing and non-expressing cells"

If the above is the sentiment of the authors, then the manuscript is not written to support this sentiment clearly, starting with this misleading sentence in the Abstract, "The boundary domain is absent in Lmx1a-deficient mice, which exhibit defects in sensory organ segregation, and is disrupted by the inhibition of ROCK-dependent actomyosin contractility."

(2) As acknowledged by the authors, the data as they currently stand could be explained by Lmx1a functioning in specifying the non-sensory fate and may not function directly in boundary formation. With this caveat in mind, the role of Lmx1a in boundary formation remains unclear.

(3) I feel like the word "orchestrate" in the title is an overstatement.

---

## [Author Response]

The following is the authors’ response to the original reviews.

**Reviewer #1 (Public review):**
Summary:This manuscript investigated the mechanism underlying boundary formation necessary for proper separation of vestibular sensory end organs. In both chick and mouse embryos, it was shown that a population of cells abutting the sensory (marked by high Sox2 expression) /nonsensory cell populations (marked by Lmx1a expression) undergo apical expansion, elongation, alignment and basal constriction to separate the lateral crista (LC) from the utricle. Using Lmx1a mouse mutant, organ cultures, pharmacological and viral-mediated Rock inhibition, it was demonstrated that the Lmx1a transcription factor and Rock-mediated actomyosin contractility is required for boundary formation and LC-utricle separation.Strengths:Overall, the morphometric analyses were done rigorously and revealed novel boundary cell behaviors. The requirement of Lmx1a and Rock activity in boundary formation was convincingly demonstrated.Weaknesses:However, the precise roles of Lmx1a and Rock in regulating cell behaviors during boundary formation were not clearly fleshed out. For example, phenotypic analysis of Lmx1a was rather cursory; it is unclear how Lmx1a, expressed in half of the boundary domain, control boundary cell behaviors and prevent cell mixing between Lmx1a+ and Lmx1a- compartments? Well-established mechanisms and molecules for boundary formation were not investigated (e.g. differential adhesion via cadherins, cell repulsion via ephrin-Eph signaling). Moreover, within the boundary domain, it is unclear whether apical multicellular rosettes and basal constrictions are drivers of boundary formation, as boundary can still form when these cell behaviors were inhibited. Involvement of other cell behaviors, such as radial cell intercalation and oriented cell division, also warrant consideration. With these lingering questions, the mechanistic advance of the present study is somewhat incremental.

We have acknowledged the lingering questions this referee points out in our Discussion and agree that the roles of differential cell adhesion and cell intercalation would be worth exploring in further studies. Despite these remaining questions, the conceptual advances are significant, since this study provides the first evidence that a tissue boundary forms in between segregating sensory organs in the inner ear (there are only a handful of embryonic tissues in which a tissue boundary has been found in vertebrates) and highlights the evolutionary conservation of this process. This work also provides a strong descriptive basis for any future study investigating the mechanisms of tissue boundary formation in the mouse and chicken embryonic inner ear.

**Reviewer #2 (Public review):**
Summary:Chen et al. describe the mechanisms that separate the common pan-sensory progenitor region into individual sensory patches, which presage the formation of the sensory epithelium in each of the inner ear organs. By focusing on the separation of the anterior and then lateral cristae, they find that long supra-cellular cables form at the interface of the pansensory domain and the forming cristae. They find that at these interfaces, the cells have a larger apical surface area, due to basal constriction, and Sox2 is down-regulated. Through analysis of Lmx1 mutants, the authors suggest that while Lmx1 is necessary for the complete segregation of the sensory organs, it is likely not necessary for the initial boundary formation, and the down-regulation of Sox2.Strengths:The manuscript adds to our knowledge and provides valuable mechanistic insight into sensory organ segregation. Of particular interest are the cell biological mechanisms: The authors show that contractility directed by ROCK is important for the maintenance of the boundary and segregation of sensory organs.Weaknesses:The manuscript would benefit from a more in-depth look at contractility - the current images of PMLC are not too convincing. Can the authors look at p or ppMLC expression in an apical view? Are they expressed in the boundary along the actin cables? Does Y-27362 inhibit this expression?The authors suggest that one role for ROCK is the basal constriction. I was a little confused about basal constriction. Are these the initial steps in the thinning of the intervening nonsensory regions between the sensory organs? What happens to the basally constricted cells as this process continues?

In our hands, the PMLC immunostaining gave a punctate staining in epithelial cells and was difficult to image and interpret in whole-mount preparations, which did not allow us to investigate its specific association to the actin-cable-like structures. It is a very valuable suggestion to try alternative methods of fixation to improve the quality of the staining and images in future work.

The basal constriction of the cells at the border of the sensory organs was not always clearly visible in freshly-fixed samples, and was absent in the majority of short-term organotypic cultures in control medium, which made it impossible to ascertain the role of ROCK in its formation using pharmacological approaches in vitro (see Figure 7 and corresponding Result section). On the other hand, the overexpression of a dominant-negative form of ROCK (RCII-GFP) in ovo using RCAS revealed a persistence of basal constriction in transfected cells despite a disorganisation of the boundary domain (Figure 8). We conclude from these experiments that ROCK activity is not necessary for the formation and maintenance of the basal constriction. We also remain uncertain about the exact role of this basal constriction. It could be either a cause or consequence of the expansion of the apical surface of cells in the boundary domain, it could contribute to the limitation of cell intermingling and the formation of the actin-cable-like structure at the interface of Lmx1a-expressing and non-expressing cells, and may indeed prefigure some of the further changes in cell morphology occurring in non-sensory domains separating the sensory organs (cell flattening and constrictions of the epithelial walls in between sensory organs).

The steps the authors explore happen after boundaries are established. This correlates with a down-regulation of Sox2, and the formation of a boundary. What is known about the expression of molecules that may underlie the apparent interfacial tension at the boundaries? Is there any evidence for differential adhesion or for Eph-Ephrin signalling? Is there a role for Notch signalling or a role for Jag1 as detailed in the group's 2017 paper?

Great questions. It is indeed likely that some form of differential cell tension and/or adhesion participates to the formation and maintenance of this boundary, and we have mentioned in the discussion some of the usual suspects (cadherins, eph/ephrin signalling,…) although it is beyond the scope of this paper to determine their roles in this context.

As we have discussed in this paper and in our 2017 study (see also Ma and Zhang, Development, 2015 Feb 15;142(4):763-73. doi: 10.1242/dev.113662) we believe that Notch signalling is maintaining prosensory character, and its down-regulation by Lmx1a/b expression is required for the specification of the non-sensory domains in between segregating sensory organs. Although we have not tested this directly in this study, any disruption in Notch signalling would be expected to affect indirectly the formation or maintenance of the boundary domain.

A comment on whether cellular intercalation/rearrangements may underlie some of the observed tissue changes.

We have not addressed this topic directly in the present study but we have included a brief comment on the potential implication of cellular intercalation and rearrangements in the discussion: “It is also possible that the repositioning of cells through medial intercalation could contribute to the straightening of the boundary as well as the widening of the nonsensory territories in between sensory patches.”

The change in the long axis appears to correlate with the expression of Lmx1a (Fig 5d). The authors could discuss this more. Are these changes associated with altered PCP/Vangl2 expression?

We are not sure about the first point raised by the referee. We have quantified cell elongation and orientation in Lmx1a-GFP heterozygous and homozygous (null) mice, and our results suggest that the elongation of the cells occurs throughout the boundary domain, and is probably not dependent on Lmx1a expression (boundary cells are in fact more elongated in the Lmx1a mutant). We have not investigated the expression of components of the planar cell polarity pathway. This is a very interesting suggestion, worth exploring in further studies.

**Reviewer #3 (Public review):**
Summary:Lmx1a is an orthologue of apterous in flies, which is important for dorsal-ventral border formation in the wing disc. Previously, this research group has described the importance of the chicken Lmx1b in establishing the boundary between sensory and non-sensory domains in the chicken inner ear. Here, the authors described a series of cellular changes during border formation in the chicken inner ear, including alignment of cells at the apical border and concomitant constriction basally. The authors extended these observations to the mouse inner ear and showed that these morphological changes occurred at the border of Lmx1a positive and negative regions, and these changes failed to develop in Lmx1a mutants. Furthermore, the authors demonstrated that the ROCK-dependent actomyosin contractility is important for this border formation and blocking ROCK function affected epithelial basal constriction and border formation in both in vitro and in vivo systems.Strengths:The morphological changes described during border formation in the developing inner ear are interesting. Linking these changes to the function of Lmx1a and ROCK dependent actomyosin contractile function are provocative.Weaknesses:There are several outstanding issues that need to be clarified before one could pin the morphological changes observed being causal to border formation and that Lmx1a and ROCK are involved.

We have addressed the specific comments and suggestions of the reviewer below. We wish however to point out that we do not think that ROCK activity is required for the formation or maintenance of the basal constriction at the interface of Lmx1a-expressing and nonexpressing cells (see previous answer to referee #2)

**Reviewer #1 (Recommendations for the authors):**
Specific comments:(1) Figures 1 and 2, and related text. Based on the whole-mount images shown, the anterior otocyst appeared to be a stratified epithelium with multiple cell layers. If so, it should be clarified whether the x-y view of in the "apical" and "basal" plane are from cells residing in the apical and basal layers, respectively. Moreover, it would be helpful to include a "stage 4", a later stage to show if and when basal constrictions resolve.

In fact, at these early stages of development, the otic epithelium is “pseudostratified”: it is formed by a single layer of irregularly shaped cells, each extending from the base to the apical aspect of the epithelium, but with their nuclei residing at distinct positions along this basal-apical axis as mitotic cells progress through the cell cycle. The nuclei divide at the surface of the epithelium, then move back to the most basal planes within daughter cells during interphase. This process, known as interkinetic nuclear migration, has been well described in the embryonic neural tube and occurs throughout the developing otic epithelium (e.g. Orr, Dev Biol. 1975, 47,325-340, Ohta et al., Dev Biol. 2010 Sep 15;347(2):369–381. doi: 10.1016/j.ydbio.2010.09.002;). Consequently, the nuclei visible in apical or basal planes in x-y views belong to cells extending from the base to the apex of the epithelium, but which are at different stages of the cell cycle.

We have not included a late stage of sensory organ segregation in this study (apart from a P0 stage in the mouse inner ear, see Figure 4) since data about later stages of sensory organ morphogenesis are available in other studies, including our Mann et al. eLife 2017 paper describing Lmx1a-GFP expression in the embryonic mouse inner ear.

(2) Related to above, the observed changes in cell organization raised the possibility that the apical multicellular rosettes and basal constrictions observed in Stage 3 (and 2) could be intermediates of radial cell intercalations, which would lead to expansion of the space between sensory organs and thinning of the boundary domains. To see if it might be happening, it would be helpful to include DAPI staining to show the overall tissue architecture at different stages and use optical reconstruction to assess the thickness of the epithelium in the presumptive boundary domain over time.

We agree with this referee. Besides cell addition by proliferation and/or changes in cell morphology, radial cell intercalations could indeed contribute to the spatial segregation of inner ear sensory organs (a brief statement on this possibility was added to the Discussion). It is clear from images shown in Figure 4 (and from other studies) that the non-sensory domain separating the cristae from the utricle gets flatter and its cells also enlarge as development proceeds. We do not think that DAPI staining is required to demonstrate this. Perhaps the best way to show that radial cell intercalations occur would be to perform liveimaging of the otic epithelium, but this is technically challenging in the mouse or chicken inner ear. An alternative model system might be the zebrafish inner ear, in which some liveimaging data have shown a progressive down-regulation of Jag1 expression during sensory organ segregation (and a flattening of “boundary domains”), suggesting a conservation of the basic mechanisms at play (Ma and Zhang, Development, 2015 Feb 15;142(4):763-73. doi: 10.1242/dev.113662).

(3) Similarly, it would be helpful to include the DAPI counterstain in Figures 4, 7, and 8 to show the overall tissue architecture.

We do not have DAPI staining for these particular images but in most cases, Sox2 immunostaining gives a decent indication of tissue morphology.

(4) Figure 2(z) and Figure 4d. The arrows pointing at the basal constrictions are obstructing the view of the basement membrane area, making it difficult to appreciate the morphological changes. They should be moved to the side. Can the authors comment whether they saw evidence for radial intercalations (e.g. thinning of the boundary domain) or partial unzippering of adjoining compartments along the basal constrictions?

The arrows in Figure 2(z) and Figure 4d have been moved to the side of the panels.

See previous comment. Besides the presence of multicellular rosettes, we have not seen direct evidence of radial cell intercalation – this would be best investigated using liveimaging. As development proceeds, the epithelial domain separating adjoining sensory organs becomes wider. The cells that compose it gradually enlarge and flatten, as can be seen for example at P0 in the mouse inner ear (Figure 4g).

(5) Figures 3 and 5, and related text. It should be clarified whether the measurements were all taken from the surface cells. For Fig. 3e and 5d, the mean alignment angles of the cell long axis in the boundary regions should be provided in the text.

The sensory epithelium in the otocyst is pseudostratified, hence, the measurement was taken from the surface of all epithelial cells labelled with F-actin.

We have added histograms representing the angular distribution of the cell long axis orientations in the boundary region to Figure 3 and Figure 5 Supplementary 1. We believe that this type of representation is more informative than the numerical value of the mean alignment angles of the cell long axis for defined sub-domains.

(6) It would be helpful to also quantify basal constrictions using the cell skeleton analysis. In addition, it would be helpful to show x-y views of cell morphology at the level of basal constrictions in the mouse tissue, similar to the chick otocyst shown in Figure 2.

The data that we have collected do not allow a precise quantification of basal constrictions with cell skeleton analysis, due to the generally fuzzy nature of F-actin staining in the basal planes of the epithelium. However, we have followed the referee’s advice and analysed Factin staining in x-y views in the Lmx1a-GFP knock-in (heterozygous) mice. We found that the first signs of basal F-actin enrichment and multicellular actin-cable like structures at the interface of Lmx1a-positive and negative cells are visible at E11.5, and F-actin staining in the basal planes increases in intensity and extent at E13.5. (shown in new Figure 4 – Supplementary Figure 1).

(7) Figure 5 and related text. It would be informative to analyze Lmx1a mutants at early stages (E11-E13) to pinpoint cell behavior defects during boundary formation.

We chose the E15 stage because it is one at which we can unequivocally recognize and easily image and analyse the boundary domain from a cytoarchitectural point of view. We recognize that it would have been worth including earlier stages in this analysis but have not been able to perform these additional studies due to time constraints and unavailability of biological material.

(8) Figure 5-Figure S1, the quantifications suggest that Lmx1a loss had both cellautonomous and non-autonomous effects on boundary cell behaviors. This is an interesting finding, and its implication should be discussed.

It is well-known that the absence of Lmx1a function induces a very complex (and variable) phenotype in terms of inner ear morphology and patterning defects. It is also clear from this study that the absence of Lmx1 causes non-cell autonomous defects in the boundary domain and we have already mentioned this in the discussion: “Finally, the patterning abnormalities in Lmx1a^GFP/GFP^ samples occurred in both GFP-positive and negative territories, which points at some type of interaction between Lmx1a-expressing and nonexpressing cells, and the possibility that the boundary domain is also a signalling centre influencing the differentiation of adjacent territories.”

(9) Figure 6 and related text. To correlate myosin II activity with boundary cell behaviors, it would be important to immunolocalize pMLC in the boundary domain in whole-mount otocyst preparations from stage 1 to stage 3.

We tried to perform the suggested immunostaining experiments, but in our hands at least, the antibody used did not produce good quality staining in whole-mount preparations. We have therefore included images of sectioned otic tissue, which show some enrichment in pMLC immunostaining at the interface of segregating organs (Figure 6).

(10) Figures 7 and 8. A caveat of long-term Rock inhibition is that it can affect cell proliferation and differentiation of both sensory and non-sensory cells, which would cause secondary effects on boundary formation. This caveat was not adequately addressed. For example, does Rock signaling control either the rate or the orientation of cell division to promote boundary formation? Together with the mild effect of acute Rock inhibition, the precise role of Rock signaling in boundary formation remains unclear.

We absolutely agree that the exact function of ROCK could not be ascertained in the in vitro experiments, for the reasons we have highlighted in the manuscript (no clear effect in short term treatments, great level of tissue disorganisation in long-term treatments). This prompted us to turn to an in ovo approach. The picture remains uncertain in relation to the role of ROCK in regulating cell division/intercalation but we have been at least able to show a requirement for the maintenance of an organized and regular boundary.

(11) Figure 8. RCII-GFP likely also have non-autonomous effects on cell apical surface area. In 8d, it would be informative to include cell area quantifications of the GFP control for comparison.

It is possible that some non-autonomous effects are produced by RCII-GFP expression, but these were not the focus of the present study and are not particularly relevant in the context of large patches of overexpression, as obtained with RCAS vectors.

We have added cell surface area quantifications of the control RCAS-GFP construct for comparison (Figure 8e).

(12) The significance of the presence of cell divisions shown in Figure 9 is unclear. It would be informative to include some additional analysis, such as (a) quantify orientation of cell divisions in and around the boundary domain and (b) determine whether patterns of cell division in the sensory and nonsensory regions are disrupted in Lmx1a mutants.

These are indeed fascinating questions, but which would require considerable work to answer and are beyond the scope of this paper.

Minor comments:(1) Figure 1. It should be clarified whether e', h' and k' are showing cortical F-actin of surface cells. Do the arrowheads in i' and l' correspond to the position of either of the arrowheads in h' and k', respectively?

The epithelium in the otocyst is pseudostratified. Therefore, images e’, h’, k’ display F-actin labelling on the surface of tissue composed of a single cell layer. We have added arrows to images e”, h”, and k” to indicate the corresponding position of z-projections and included appropriate explanation in the legend of Figure 1: “Black arrows on the side of images e”, h”, and k” indicate the corresponding position of z-projections.”

(2) Figure 3-Figure S1. Please mark the orientation of the images shown.

We labelled the sensory organs in the figure to allow for recognizing the orientation.

(3) Figure 4. Orthogonal reconstructions should be labeled (z) to be consistent with other figures.

We have corrected the labelling in the orthogonal reconstruction to (z).

(4) Figure 4g. It is not clear what is in the dark area between the two bands of Lmx1a+ cells next to the utricle and the LC. Are those cells Lmx1a negative? It is unclear whether a second boundary domain formed or the original boundary domain split into two between E15 and P0? Showing the E15 control tissue from Figure 5 would be more informative than P0.

In this particular sample there seems to be a folding of the tissue (visible in z-reconstructions) that could affect the appearance of the projection shown in 4g. We believe the P0 is a valuable addition to the E15 data, showing a slightly later stage in the development of the vestibular organs.

(5) Figure 5a, e. Magnified regions shown in b and f should be boxed correspondingly.

This figure has been revised. We realized that the previous low-magnification shown in (e) (now h) was from a different sample than the one shown in the high-magnification view. The new figure now includes the right low-magnification sample (in h) and the regions shown in the high-magnification views have been boxed.

(6) Figure 8f, h, j. Magnified regions shown in g, i and k should be boxed correspondingly.

The magnified regions were boxed in Figure 8 f, h, and j. Additionally, black arrows have been placed next to images 8g", 8i", and 8k" to highlight the positions of the z-projections. An appropriate explanation has also been added to the figure legend.

(9) Figure 8. It would be helpful to show merged images of GFP and F-actin, to better appreciate cell morphology of GFP+ and GFP- cells.

As requested, we have added images showing overlap of GFP and F-actin channels in Figure 8.

**Reviewer #2 (Recommendations for the authors):**
The PMLC staining could be improved. Two decent antibodies are the p-MLC and pp-MLC antibodies from CST. pp-MLC works very well after TCA fixation as detailed in https://www.researchsquare.com/article/rs-2508957/latest . As phalloidin does not work well after TCA fixation, affadin works very well for segmenting cells.If the authors do not wish to repeat the pMLC staining, the details of the antibody used should be mentioned.

We used mouse IgG1 Phospho-Myosin Light Chain 2 (Ser19) from Cell Signaling Technology (catalogue number #3675) in our immunohistochemistry for PMLC. This is one of the two antibodies recommended by the reviewer #2. Information about this antibody has now been included in material and methods. This antibody has been referenced by many manuscripts, but unfortunately, in our hands at least, it did not perform well in whole-mount preparations.

A statement on the availability of the data should be included.

We have included a statement on the data availability: “All data generated or analysed during this study is available upon request.”

**Reviewer #3 (Recommendations for the authors):**
Outstanding issues:(1) Morphological description: The apical alignment of epithelial cells at the border is clear but not the upward pull of the basal lamina. Very often, it seems to be the Sox2 staining that shows the upward pull better than the F-actin staining. Perhaps, adding an anti-laminin staining to indicate the basement membrane may help.

Indeed, the upward pull of the basement membrane is not always very clear. We performed some anti-laminin immunostaining on mouse cryosections and provide below (Figure 1) an example of such experiment. The results appear to confirm an upward displacement of the basement membrane in the region separating the lateral crista from the utricle in the E13 mouse inner ear, but given the preliminary nature of these experiments, we believe that these results do not warrant inclusion in the manuscript. The term “pull” is somehow implying that the epithelial cells are responsible for the upward movement of the basement membrane, but since we do not have direct evidence that this is the case, we have replaced “pull” by “displacement” throughout the text.

(2) It is not clear how well the cellular changes are correlated with the timing of border formation as some of the ages shown in the study seem to be well after the sensory patches were separated and the border was established.

For some experiments (for example E15 in the comparison of mouse Lmx1a-GFP heterozygous and homozygous inner ear tissue; E6 for the RCAS experiments), the early stages of boundary formation are not covered because we decided to focus our analysis on the late consequences of manipulating Lmx1a/ROCK activity in terms of sensory organ segregation. The dataset is more comprehensive for the control developmental series in the chicken and mouse inner ear.

(3) The Lmx1a data, as they currently stand could be explained by Lmx1a being required for non-sensory development and not necessarily border formation. Additionally, the relationship between ROCK and Lmx1a was not investigated. Since the investigators have established the molecular mechanisms of Lmx1 function using the chicken system previously, the authors could try to correlate the morphological events described here with the molecular evidence for Lmx1 functioning during border formation in the same chicken system. Right now, only the expression of Sox2 is used to correlate with the cellular events, and not Lmx1, Jag1 or notch.

These are valid points. Exploring in detail the epistatic relationships between Notch signalling/Lmx1a/ROCK/boundary formation in the chicken model would be indeed very interesting but would require extensive work using both gain and loss-of-function approaches, combined with the analysis of multiple markers (Jag1/Sox2/Lmx1b/PMLC/Factin..). At this point, and in agreement with the referee’s comment, we believe that Lmx1a is above all required for the adoption of the non-sensory fate. The loss of Lmx1a function in the mouse inner ear produce defects in the patterning and cellular features of the boundary domain, but these may be late consequences of the abnormal differentiation of the nonsensory domains that separate sensory organs. Furthermore, ROCK activity does not appear to be required for Sox2 expression (i.e. adoption or maintenance of the sensory fate) since the overexpression of RCII-GFP does not prevent Sox2 expression in the chicken inner ear. This fits with a model in which Notch/Lmx1a regulate cell differentiation whilst ROCK acts independently or downstream of these factors during boundary formation.

Specific comments:(1) Figure 1. The downregulation of Sox2 is consistent between panels h and k, but not between panels e and h. The orthogonal sections showing basal constriction in h' and k' are not clear.

The downregulation is noticeable along the lower edge of the crista shown in h; the region selected for the high-magnification view sits at an intermediate level of segregation (and Sox2 downregulation).

The basal constriction is not very clear in h, but becomes easier to visualize in k. We have displaced the arrow pointing at the constriction, which hopefully helps.

(2) Figure 2. Where was the Z axis taken from? One seems to be able to imagine the basal constriction better in the anti-Sox2 panel than the F-actin panel. A stain outlining the basement membrane better could help.

Arrows have been added on the side of the horizontal views to mark the location of the zreconstruction. See our previous replies to comments addressing the upward displacement of the basement membrane.

(3) Figure 4I question the ROI being chosen in this figure, which seems to be in the middle of a triad between LC, prosensory/utricle and the AC, rather than between AC and LC. If so, please revise the title of the figure. This could also account for the better evidence of the apical alignment in the upper part of the f panel.

We have corrected the text.

In this figure, the basal constriction is a little clearer in the orthogonal cuts, but it is not clear where these sections were taken from.

We have added black arrows next to images 4c’, 4f’, and 4i’ to indicate the positions of the zprojections.

By E13.5, the LC is a separate entity from the utricle, it makes one wonder how well the basal constriction is correlated with border formation. The apical alignment is also present by P0, which raises the question that the apical alignment and basal restriction may be more correlated with differentiation of non-sensory tissue rather than associated with border formation.

We agree E13.5 is a relatively late stage, and the basal constriction was not always very pronounced. The new data included in the revised version include images of basal planes of the boundary domain at E11.5, which reveal F-actin enrichment and the formation of an actin-cable-like structure (Figure 4 suppl. Fig1). Furthermore, the chicken dataset shows that the changes in cell size, alignment, and the formation of actin-cable-like structure precede sensory patch segregation and are visible when Sox2 expression starts to be downregulated in prospective non-sensory tissue (Figure 1, Figure 2). Considering the results from both species, we conclude that these localised cellular changes occur relatively early in the sequence of events leading to sensory patch segregation, as opposed to being a late consequence of the differentiation of the non-sensory territories.

I don't follow the (x) cuts for panels h and I, as to where they were taken from and why there seems to be an epithelial curvature and what it was supposed to represent.

We have added black arrows next to the panels 4c’, 4f’, and 4i’ to indicate the positions of the z-projections and modified the legend accordingly. The epithelial curvature is probably due to the folding of the tissue bordering the sensory organs during the manipulation/mounting of the tissue for imaging.

(4) Figure 5 The control images do not show the apical alignment and the basal constriction well. This could be because of the age of choice, E15, was a little late. Unfortunately, the unclarity of the control results makes it difficult for illustrating the lack of cellular changes in the mutant. The only take-home message that one could extract from this figure is a mild mixing of Sox2 and Lmx1a-Gfp cells in the mutant and not much else. Also, please indicate the level where (x) was taken from.

Black arrows have been placed next to images 5e and 5l to highlight the positions of the zprojections. The stage E15 chosen for analysis was appropriate to compare the boundary domains once segregation is normally completed. We believe the results show some differences in the cellular features of the boundary domain in the Lmx1a-null mouse, and we have in fact quantified this using Epitool in Figure 5 – Suppl. Fig 1. Cells are more elongated and better aligned in the Lmx1a-null than in the heterozygous samples.

(5) Figure 7. I think the cellular disruption caused by the ROCK inhibitor, shown in q', is too severe to be able to pin to a specific effect of ROCK on border formation. In that regard, the ectopic expression of the dominant negative form of ROCK using RCAS approach is better, even though because it is a replication competent form of RCAS, it is still difficult to correlate infected cells to functional disruption.

We used a replication-competent construct to induce a large patch of infection, increasing our chances of observing a defect in sensory organ segregation and boundary formation. We agree that this approach does not allow us to control the timing of overexpression, but the mosaicism in gene expression, allowing us to compare in the same tissue large regions with/without perturbed ROCK activity, proved more informative than the pharmacological/in vitro experiments.

(6) Figure 8. Outline the ROI of i in h, and k in j. Outline in k the comparable region in k'. In k", F-actin staining is not uniform. Indicate where (x) was taken from in K.

The magnified regions were boxed in Figure 8 f, h, and j. Region outlined in figures k’-k” has also been outlined in corresponding region in figure k. Additionally, black arrows have been placed next to images 8g", 8i", and 8k" to highlight the positions of the z-projections. An appropriate explanation has also been added to the figure legend.

Minor comments:(1) P.18, 1st paragraph, extra bracket at the end of the paragraph.

Bracket removed

(2) P.22, line 11, in ovo may be better than in vivo in this case.

We agree, this has been corrected.

(3) P.25, be consistent whether it is GFP or EGFP.

Corrected to GFP.

(4) P.26, line 5. Typo on "an"

Corrected to “and”

**Author response image 1. sa3fig1:** Expression of Laminin and Sox2 in the E13 mouse inner ear. (a-a’’’) Low magnification view of the utricle, the lateral crista, and the non-sensory (Sox2-negative) domain separating these. Laminin staining is detected at relatively high levels in the basement membrane underneath the sensory patches. At higher magnification (b-b’’’), an upward displacement of the basement membrane (arrow) is visible in the region of reduced Sox2 expression, corresponding to the “boundary domain” (bracket).